# Distinct organization of two cortico-cortical feedback pathways

Shan Shen[1,2,8], Xiaolong Jiang [1,2,3,8], Federico Scala[1,2], Jiakun Fu[1,2],
Paul Fahey [1,2], Dmitry Kobak [4], Zhenghuan Tan[1,2], Na Zhou[1,2], Jacob Reimer[1,2],
Fabian Sinz[1,2,5,6] & Andreas S. Tolias [1,2,7] ✉

Neocortical feedback is critical for attention, prediction, and learning. To mechanically understand its function requires deciphering its cell-type wiring. Recent studies revealed that feedback between primary motor to primary somatosensory areas in mice is disinhibitory, targeting vasoactive intestinal peptide-expressing interneurons, in addition to pyramidal cells. It is unknown whether this circuit motif represents a general cortico-cortical feedback organizing principle. Here we show that in contrast to this wiring rule, feedback between higher-order lateromedial visual area to primary visual cortex preferentially activates somatostatin-expressing interneurons. Functionally, both feedback circuits temporally sharpen feed-forward excitation eliciting a transient increase–followed by a prolonged decrease–in pyramidal cell activity under sustained feed-forward input. However, under feed-forward transient input, the primary motor to primary somatosensory cortex feedback facilitates bursting while lateromedial area to primary visual cortex feedback increases time precision. Our findings argue for multiple cortico-cortical feedback motifs implementing different dynamic non-linear operations.

The mammalian neocortex is composed of hierarchically organized areas, where cortical areas are reciprocally connected with both feed-forward and top-down feedback projections. Feedback connections are believed to participate in many important brain functions, such as attention[1–3], prediction[4–8], and shaping activity based on context[9–13]. Despite the importance of cortical feedback, its precise cell-type wiring logic is still not fully understood, preventing a mechanistic understanding of its modulatory role in cortical processing.

The mammalian sensory neocortex is organized in a six-layer structure, composed of distinct cell types wired in canonical circuit motifs. For example, inhibitory interneurons in cortical circuits are grouped into highly heterogeneous transcriptomic and morphological cell classes, hypothesized to exert distinct functional roles[14–17]. Therefore, it is critical to decipher the cell-type-specific wiring of

feedback connections. Among GABAergic interneurons in mouse neocortex, parvalbumin (PV), somatostatin (SOM), and vasointestinal peptide (VIP) expressing interneurons are the three main non-overlapping cell classes that comprise more than 80% of the GABAergic interneurons[15,18–20]. Importantly, these cell classes follow specific rules governing their connections with other interneuron types and local excitatory neurons[14,19,21]. Specifically, a key connectivity rule is that VIP+ interneurons preferentially inhibit SOM+ interneurons which in turn inhibit local pyramidal cells[14,21,22]. Recent studies indicate that feedback projections primarily recruit this disinhibitory disynaptic circuit by preferentially activating VIP+ interneurons[23–26], supporting the view that the primary effect of feedback is a disinhibition of the target areas[27]. However, the cortical feedback projections to interneurons studied so far mainly

[1]Center for Neuroscience and Artificial Intelligence, Baylor College of Medicine, Houston, TX, USA. [2]Department of Neuroscience, Baylor College of Medicine, Houston, TX, USA. [3]Jan and Dan Duncan Neurological Research Institute at Texas Children's Hospital, Houston, TX, USA. [4]Institute for Ophthalmic Research, University of Tübingen, Tübingen, Germany. [5]Bernstein Center for Computational Neuroscience, University of Tübingen, Tübingen, Germany. [6]Institute for Bioinformatics and Medical Informatics, University of Tübingen, Tübingen, Germany. [7]Department of Electrical and Computational Engineering, Rice University, Houston, TX, USA. [8]These authors contributed equally: Shan Shen, Xiaolong Jiang. ✉e-mail: astolias@bcm.edu

focused on connections between brain regions across different modalities such as between motor and sensory areas[22,24,26]. Therefore, it is not clear if feedback projections recruit the same disinhibitory circuit between hierarchically organized areas within the same sensory modality and whether the feedback-eliciting disinhibitory circuit motif is a universal organizing principle of cortical feedback.

The mouse visual cortex is an ideal model to address this question given its hierarchically organized extrastriate areas[28–36]. Here we focused on feedback connections from the lateral-medial area (LM), which is believed to be analogous to area V2 in primates[33], to area V1. We compared the organization of this within-hierarchy feedback to the previously studied feedback pathways across modalities, from the vibrissal primary motor cortex (vM1) to vibrissal S1 (vS1), to examine if there is a general rule governing the organization of feedback pathways across these two different pathways. We found major differences in both the cell-type-specific wiring rules and the functional impact of feedback projections between these two pathways. LM to V1 feedback connected more strongly to SOM+ cells than to VIP+ cells while vM1 to vS1 feedback showed the opposite pattern, consistent with previous studies[23]. When paired with sustained positive current injection into the cell bodies of pyramidal cells in either V1 or vS1, feedback projections had a similar effect on the activity of pyramidal cells: activation of feedback temporally sharpened the feed-forward excitation by eliciting a transient increase followed by a sustained decrease in firing rate. However, when paired with a brief positive feed-forward current pulse, vM1 to vS1 feedback facilitated bursting of Layer 5 (L5) intrinsically bursty (IB) cells. In contrast, under the same brief feed-forward input, LM to V1 feedback increased the probability of a second spike but eliminated subsequent spikes, in agreement with temporal sharpening. Our results argue for multiple feedback circuit motifs specialized for distinct dynamic non-linear operations.

## Results
### Distribution of feedback axon terminals
To study the projection pattern and connectivity of feedback pathways, we injected the adeno-associated virus (AAV2/1) expressing channelrhodopsin-2 (ChR2)-YFP in either LM or vM1 (Fig. 1a, Methods), covering all cortical layers (Methods, Supplementary Fig. S1). LM was identified with intrinsic optical imaging[28,37] (Fig. 1b), while vM1 was identified stereotaxically (0.9 mm lateral and 1.1 mm anterior of bregma[23,38] (Fig. 1a, see Methods).

Two to four weeks later, we used two-photon microscopy to image the fluorescence within a 3 mm cranial window centered on either V1 (centered at 3.0 mm lateral, 1.5 mm anterior to lambda, Fig. 1a, red dashed circle, and Fig. 1c) or vS1 (centered at 3.5 mm lateral and 1.5 mm posterior to bregma, Fig. 1a, blue dashed circle, and Fig. 1d). Consistent with a recent study[39], feedback projections from LM mainly targeted the retinotopically matched area in V1 (Fig. 1c and d; z-score of fluorescence in retinotopically matched and unmatched areas in V1; $p = 0.008$, two-sided Wilcoxon signed-rank test, $n = 8$ animals), confirming a spatial specificity in the LM to V1 feedback projections[39]. vM1 to vS1 projections targeted vS1 and other areas such as secondary somatosensory area and posterior parietal cortex (Fig. 1d). Although we do not have direct evidence in the current study, it is possible that vM1 to vS1 projections are also organized in a topographical manner, as suggested in Mao et. al., 2011[38].

Coronal slices revealed that LM to V1 axon terminals spanned all layers, with denser projections in L1 and deep layers (L5 and L6), and sparser projections to the layers in between (Fig. 1e, left). vM1 to vS1 axon terminals were concentrated in L1 and deep layers (L5 and L6), and were very sparse in L2/3 and L4 (Fig. 1e, right), consistent with previous reports[38].

### Cell-type wiring logic of the two feedback pathways
To identify the cellular targets of both feedback pathways, we performed multi-cell simultaneous whole-cell recordings in acute brain slices prepared from either V1 or vS1 areas (Fig. 2a, see Methods). We recorded from excitatory cells and the major genetically identified classes of interneurons in Layers 1, 2/3, 4, and 5 ($n = 516$ cells in total). Distinct interneuronal classes were identified using different mouse lines (PV-Cre/Ai9, SOM-Cre/Ai9, or VIP-Cre/Ai9) and were further confirmed by *post hoc* analyses of their morphological and electrophysiological properties (see Methods, Supplementary Figs. S2, S3). In particular, we excluded the activities of fast-spiking cells from the SOM+ group in the analysis, because of their distinct firing pattern and morphological features[14,40,41] (refer to Supplementary Fig. S2 for detailed descriptions). We recorded excitatory postsynaptic currents (EPSCs) and excitatory postsynaptic potentials (EPSPs) from each neuron evoked by photostimulation of ChR2-YFP expressed in the axon terminals of feedback projections from LM or vM1 (2 ms, 470 nm; Fig. 2b–e for EPSP examples). In control experiments on animals without virus injections, we observed no response to the LED stimulation (Supplementary Fig S4).

In both V1 and vS1, we found a high proportion of cells responsive to feedback stimulation across most cortical layers and cell classes (overall 83.7%, 432/516), except for L4 in vS1, where a very low number of excitatory cells (1/13), none of the SOM+ cells (0/9) and only about half (8/14) of the PV+ cells were responsive with a latency smaller than 7 ms (Fig. 3a). The 7 ms threshold we used to define monosynaptic projections as a latency cutoff is longer than typical latencies in the literature[23]. This is because our recordings were performed under room temperature instead of physiological temperature, which preserves the quality of adult tissues[42,43] but leads to larger latencies in synaptic events[44]. We also analyzed the results with 4 ms as the latency cutoff and achieved similar results (Supplementary Fig. S6; see Methods with more detailed criteria for "responsiveness" and Supplementary Fig. S5 about latency cutoff selection).

We performed a subset of experiments (32 V1 cells out of 291 cells in total; 27 vS1 cells out of 225 cells in total) in the presence of tetrodotoxin (TTX) and 4-aminopyridine (4-AP)[45,46] and confirmed that the majority of the evoked events were monosynaptic (Methods, Supplementary Fig. S5). Consistently, we never found a pyramidal cell in any layer of either area firing in response to the feedback activation alone (Fig. 3d). Since polysynaptic events could only be elicited if local excitatory neurons are driven to fire, these results also suggest that most of the recorded feedback-evoked events were monosynaptic.

Although most of the recorded cells were responsive to feedback activation, the strength of connections varied considerably, across both the cell classes and feedback pathways (Fig. 2b–d). To compare the strength of responses to feedback across layers and cell types recorded in different brain slices and animals, we normalized the amplitudes of EPSCs and EPSPs to the mean amplitudes of the L2/3 pyramidal cells recorded from each slice, similar to approaches in previous studies[23,26], and reported the log2 normalized amplitudes (Fig. 3b, c). We found that for both feedback pathways, the normalized EPSPs and EPSCs varied considerably across different cell classes (Fig. 3b, c, $p < 10^{-6}$ for each of the four separate Kruskal–Wallis tests for EPSPs and EPSCs in V1 and vS1).

To quantify these differences, we performed statistical comparisons between normalized EPSPs and EPSCs in all pairs of neural types within each feedback pathway ($55 \times 4 = 220$ comparisons, see Methods for details, $p$ values in Supplementary Tables S1–S4), and also compared the responses of the same types of neurons across the two pathways ($10 \times 2 = 20$ comparisons, see Methods for details, $p$ values in Supplementary Table S5). These comparisons revealed two major differences in the wiring logic of the two feedback pathways.

First, we found major differences in the layer-specificity of feedback connections across the two pathways, especially in L2 to L5. In V1,

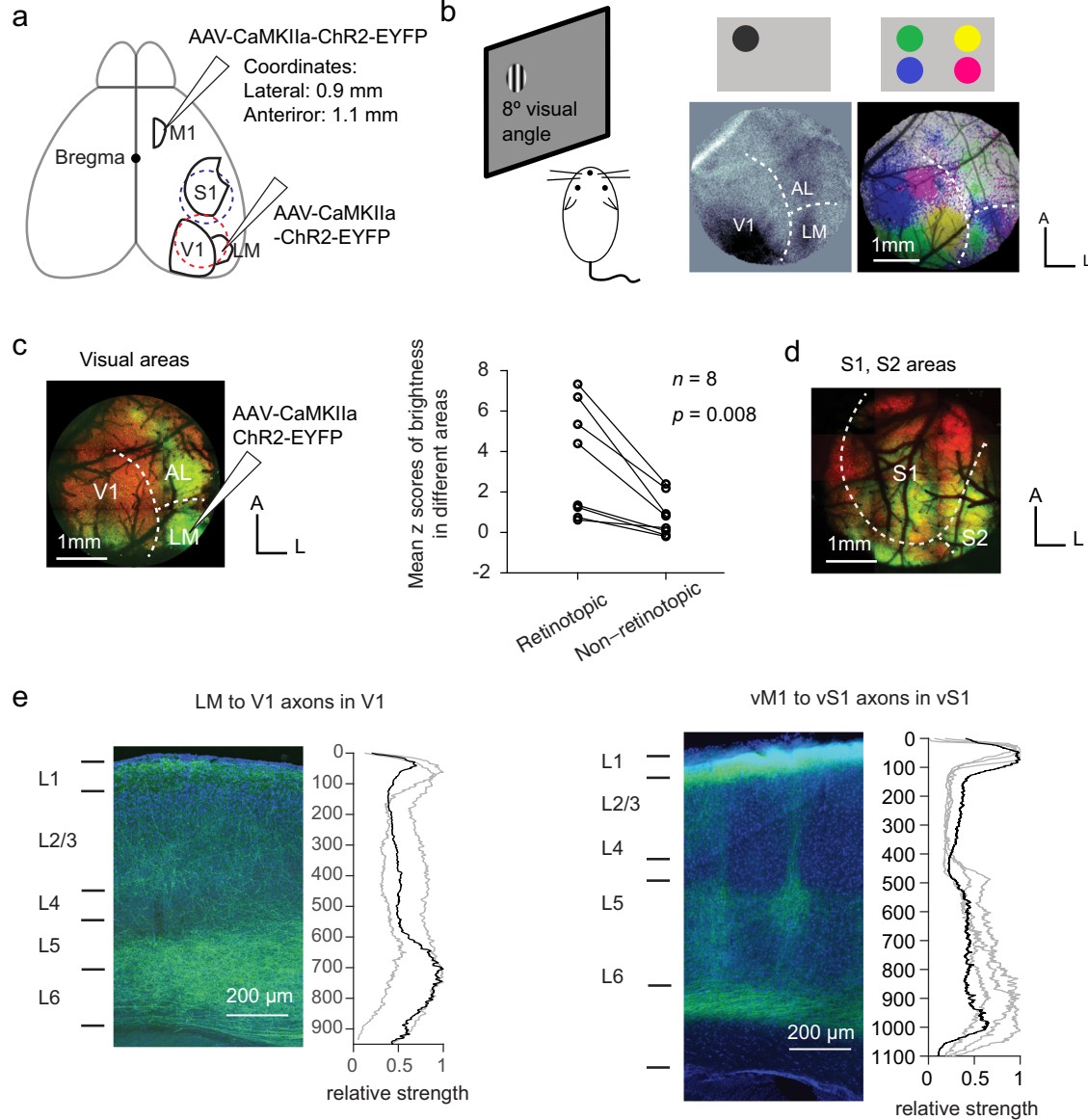

**Fig. 1 | Feedback axon terminal distribution of the two feedback pathways.**
**a** Anatomy of the areas V1, LM, vS1, and vM1. The virus was injected in either vM1 or LM in different animals. Feedback axon terminals were labeled with ChR2-YFP in either V1 (red dashed circle) or S1 (blue dashed circle). **b** Intrinsic imaging to identify V1 and LM. Left: experimental paradigm. Grating stimuli drifting horizontally or vertically were shown to the left eye, on one of the four locations on the monitor. A CCD camera was used to record brain intrinsic activity from a craniotomy exposing visual areas of the right hemisphere. Middle: intrinsic imaging map of the stimulus in the top lateral corner. The white dashed lines mark the borders among V1, LM, and AL. Right: intrinsic imaging map of stimuli in all four locations. Different colors represent brain areas responsive to stimuli in different locations (green: top-lateral; yellow: top-medial; blue: bottom-lateral; magenta: bottom-medial). Scale bar: 1 mm. **c** LM to V1 feedback projections target the retinotopic corresponding area in V1. Left: EYFP expression of the animal in (b), with

virus injected in the LM area responsive to top lateral stimulus (green) and the axon terminals mainly targeted the "green location" in V1. Right: mean z scores of the fluorescence for both retinotopically corresponding areas and non-corresponding areas in V1, for 8 animals. Fluorescence strength of the retinotopic area was significantly higher ($p = 0.008$, $n = 8$ animals, two-sided Wilcoxon signed-rank test). **d** Dorsal view of vM1 to vS1 feedback. Axon terminals were widely distributed in the somatosensory areas. **e** Laminar distribution of LM to V1 (left) or vM1 to vS1 (right) axon terminals. Left: the fluorescent image of a coronal slice in V1 or vS1, with DAPI signal in blue and EYFP signal in green. Right: Relative strength of fluorescence (averaged over 100μm horizontally) as a function of depth, normalized to the maximum (3 animals for LM to V1, and 4 animals for vM1 to vS1). The black line indicates the axon distribution corresponding to the image on the left, and the gray lines indicate other slices. Source data are provided as a Source Data file.

responses (either EPSC or EPSP) elicited in pyramidal cells, PV+ cells, and SOM+ cells in L2/3 were higher than those of their counterparts in L5 (Fig. 3b, c, for *p* values, refer to Supplementary Tables S1-S4). In contrast, in vS1, the responses of pyramidal cells and PV+ cells in L2/3 were lower than their counterparts in L5, but this was the opposite for SOM+ cells (Fig. 3b, c, for *p* values, refer to Supplementary Tables S1–S4). The response elicited by feedback in L4 cells was also different across the two feedback pathways. Specifically, in V1, the three cell types in L4 exhibited low but reliable responses to feedback

activations (Fig. 3a–c): 13/15 excitatory cells, 10/11 PV+ cells, and 9/9 SOM+ cells (Fig. 3a–c). In vS1, however, only PV+ cells were responsive to feedback activation in L4 (Fig. 3a–c). The only similarity in layer specificity we found between the two pathways was that L1 interneurons had consistently higher responses to feedback stimulation than L2/3 pyramidal cells (Fig. 3b, c), with median log2 normalized EPSPs of 0.88 in V1 and 1.1 in vS1 (Fig. 3c, Supplementary Tables S1-S4 for statistical tests). These results indicate a major difference in the layer specificity of the two feedback pathways, and are consistent

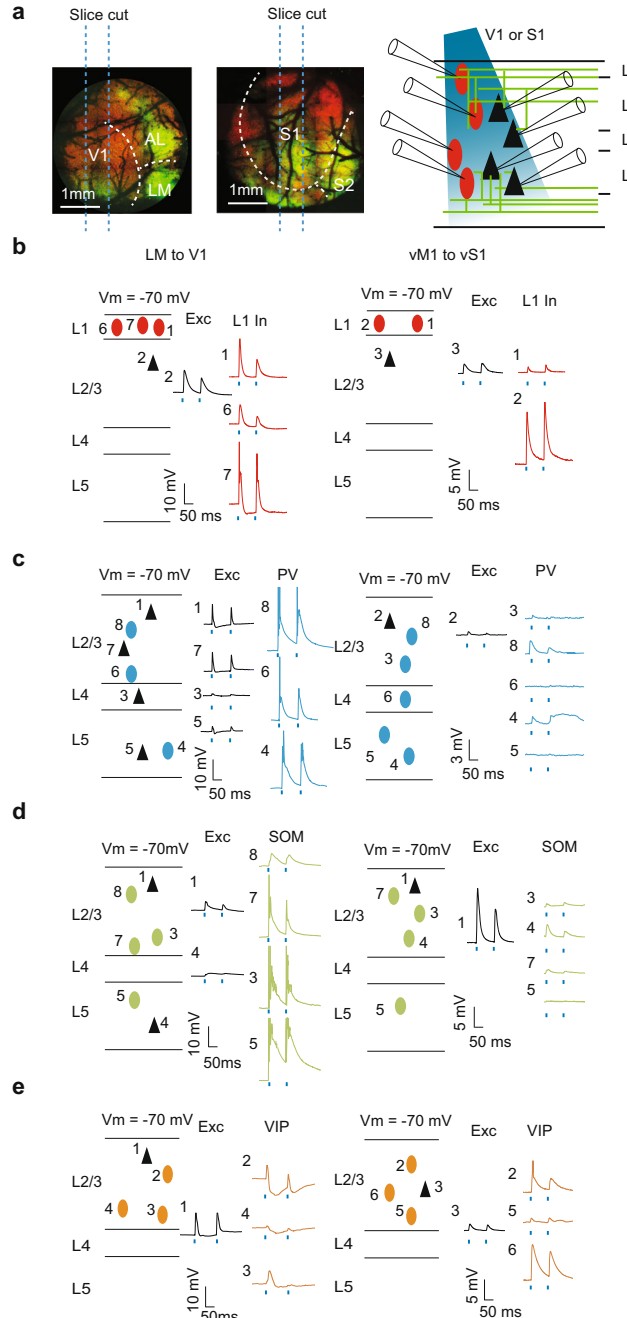

**Fig. 2 | Example slices for connectivity. a** Experimental paradigm. Left: Parasagittal slices containing V1 or vS1 Right: LM or vM1 were excluded from the slices, while the axon terminals from LM or vM1 were included in the slices. **b** Example slice recording of L1 interneurons (red ovals) and pyramidal cells (black triangles) in either V1 (left) or vS1 (right). Numbers refer to the channel number of the recording system. The blue dots mark the 2 ms blue LED pulses. **c** Example slice recording of PV+ interneurons (blue ovals) and pyramidal cells (black triangles) in either V1 (left) or vS1 (right). **d** Example slice recording of SOM+ interneurons (green ovals) and pyramidal cells (black triangles) in either V1 (left) or vS1 (right). **e** Example slice recording of L2/3 VIP+ interneurons (orange ovals) and pyramidal cells (black triangles) in either V1 (left) or vS1 (right). Source data are provided as a Source Data file.

with what we found in the laminar distribution of projection patterns (Fig. 1e).

Second, in addition to these layer-specific differences we also found a substantial difference in the overall influence of feedback on SOM+ neurons and VIP+ neurons. SOM+ responses to feedback in V1 were significantly higher than SOM+ responses in vS1 ($p < 0.02$,

Fig. 3b, c, Supplementary Table S5). In fact, vM1 feedback did not elicit spikes in any of the SOM+ neurons we recorded in vS1 (0/48, Fig. 3d), while in V1, feedback activation elicited spiking activity in a substantial fraction of SOM+ neurons (17/54). In contrast, feedback responses of VIP+ cells in L2/3 of V1, were lower than their counterparts in L2/3 of vS1 ($p < 0.01$, Fig. 3b, c, Supplementary Table S5). Moreover, a direct comparison of the normalized responses between SOM+ cells and VIP+ cells within each area showed consistent results. vM1 to vS1 feedback strongly targeted L2/3 VIP+ cells but had weaker connections to L2/3 SOM+ cells ($p = 0.007$ for EPSP, two-sided Conover test, Supplementary Table S4), and very few connections to SOM+ cells in other layers, consistent with previous reports[23]. In contrast, LM to V1 feedback strongly targeted L2/3 SOM+ cells but had significantly weaker input to L2/3 VIP+ cells ($p < 0.01$ for both EPSC and EPSP, two-sided Conover test, Supplementary Tables S1 and S2). These results indicate an opposite behavior of SOM+ cells and VIP+ cells across these two feedback pathways.

In summary, we found that the two feedback pathways differ mainly in two aspects, their layer specificity and the strength of their influence on SOM+ cells and VIP+ cells: LM to V1 feedback in general favored L2/3 over L5, while vM1 to vS1 was the opposite (with SOM+ cells being the only exception). LM to V1 feedback connection favored SOM+ cells over VIP+ cells, while vM1 to vS1 more strongly drove VIP+ cells than SOM+ cells.

The above analysis was performed using L2/3 excitatory responses as the baseline to account for inter-animal and inter-slice variability (e. g. due to variability in the amount and titer of the injected virus). The non-normalized (raw) responses of the L2/3 excitatory cells were lower in vS1 than in V1 (Supplementary Fig. S7). Since on average we injected a similar amount of virus into both areas, this might reflect a difference in the overall connectivity strength between the two feedback pathways. This possible difference, however, does not affect our main conclusions. The raw responses of SOM+ and VIP+ neurons showed the same pattern as the normalized responses (Supplementary Fig. S7). In particular, we found that that the raw EPSC values in VIP+ cells were similar between V1 and vS1, but raw EPSP values were higher in vS1 than in V1 ($p = 0.0040$, two-sided Mann Whitney $U$ test, $p$ value adjusted with Benjamin-Hochberg method). This is because the input resistance of VIP+ cells in vS1 (median 287 MΩ) is higher than that of VIP+ cells in V1 (median 204 MΩ, Supplementary Fig. S8, $p = 1.8 \times 10^{-5}$, two-sided Mann Whitney U test). With a similar amount of synaptic current, VIP+ cells in vS1 responded with a higher EPSP.

### Feedback projections in both pathways temporally sharpened the firing pattern of V1/vS1 excitatory cells, when combined with sustained feed-forward input

Next, we investigated the function of feedback projections by studying how they modulate the activity of recipient excitatory neurons in V1 or vS1, which are the principal output cells representing visual or tactile information. We first examined how feedback activation interacted with tonic depolarizing current injection designed to mimic sustained feed-forward input.

We injected 300 ms positive current steps in whole-cell configuration to the cell bodies of excitatory cells in layers 2-5 that drove trains of action potentials in these cells. We paired the depolarizing current injection with a 20 ms light pulse to activate the feedback axon terminals (Fig. 4a), randomizing the timing of the light stimulation between 100 and 200 ms after the onset of the current injection. LED-off trials were interleaved with LED-on trials (Fig. 4a).

These experiments revealed a consistent effect of feedback modulation in L2/3 and L5 for both feedback pathways: LED stimulation reliably elicited a spike within 10 ms after the LED onset, and suppressed subsequent spikes even beyond the 20 ms illumination period (Fig. 4a, for more examples, refer to Supplementary Figs. S9 and S10). Compared to LED-off trials, the firing rate of neurons during LED-on

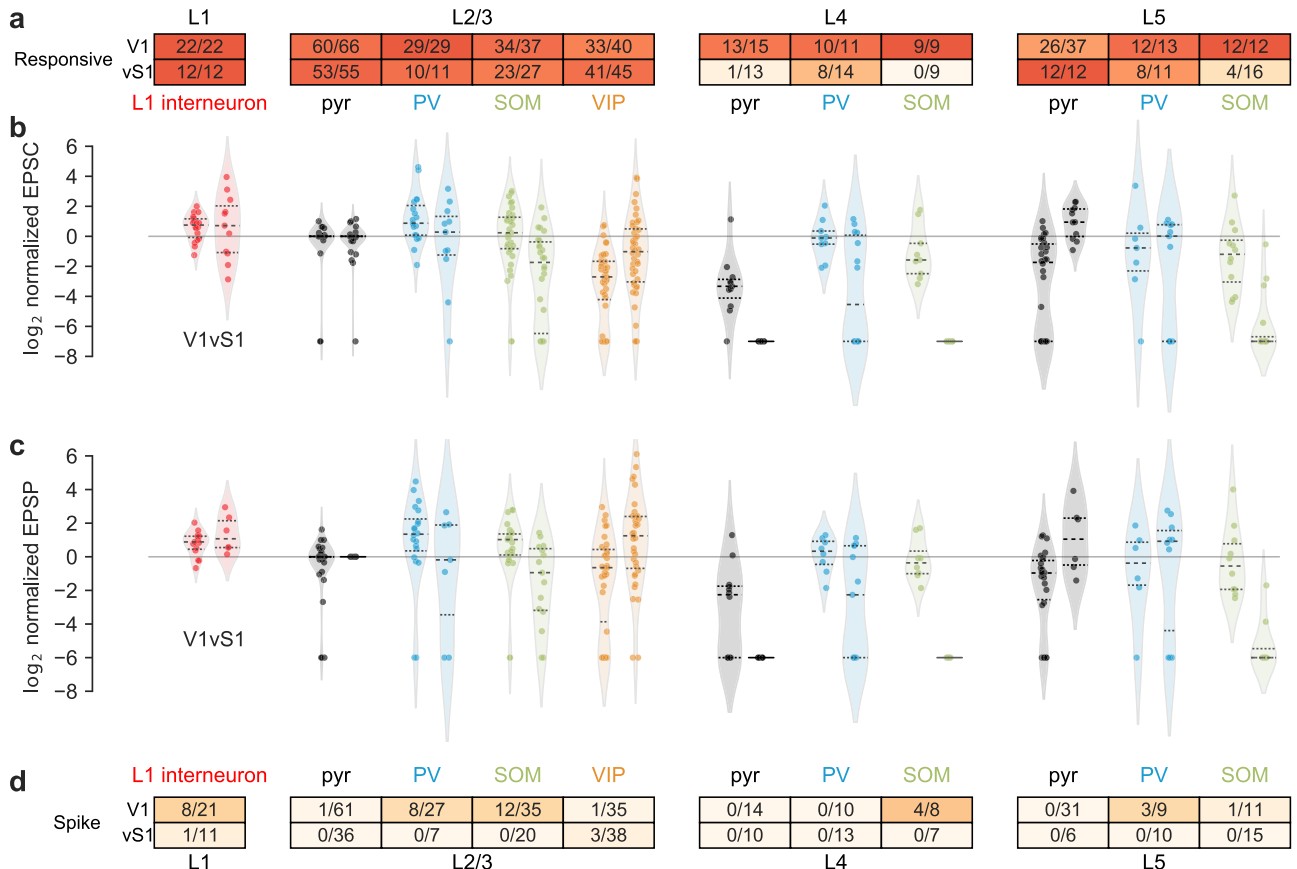

**Fig. 3 | Summary of activities in V1 and vS1 in response to feedback excitation.**
**a** The proportion of responsive cells in V1 or vS1 in different layers (number of responsive cells/total number of recorded cells). The color in the table indicates the probability level, same for panel (**d**). **b** The log2 normalized EPSC of cells in either V1 (left) or vS1 (right) normalized to the average EPSC of L2/3 pyramidal cells. Colors and positions of the violin plot indicate the cell type corresponding to the charts in panels (**a**) and (**d**). Each dot indicates one recorded cell and the color indicates the cell type. The black dashed lines indicate the quartiles (top and bottom) and the median (middle). The outlines indicate the distributions of the log2 normalized EPSCs. **c** Same as (**b**), for log2 normalized EPSP. **d** The proportion of spiking cells in V1 or vS1 in different layers (number of spiking cells/total cells recorded in the current-clamp mode). Source data are provided as a Source Data file.

trials displayed a sharp peak within 10 ms after the LED onset followed by a decrease below the baseline due to the delay of subsequent spiking (Fig. 4b). To quantify the effect during the excitation, we compared the firing rate of LED-on and -off trials within 10 ms relative to the LED onset (Fig. 4c). To quantify the subsequent delay in spiking after the feedback stimulation, we examined the time delay of spikes after stimulation offset (or the matching time points in the LED-off trials). Optogenetic feedback stimulation caused a 2- to 4-fold increase in initial firing (within the first 10 ms following LED-on) in both L2/3 and L5 in both V1 and vS1 (Fig. 4c, two-sided Wilcoxon signed-rank test, $p < =0.0099$, $n > =11$ neurons for all four comparisons), as well as an increased delay of 1.2–2 folds in the appearance of the first spike following stimulation offset (Fig. 4d, $p < =0.035$, $n > =11$ neurons for all four comparisons). Following the terminology of previous work on a different circuit[47], we call this effect "temporal sharpening". This effect was weaker in L4 neurons in V1, with a non-significant increase in the initial firing rate (Fig. 4c left, $p = 0.07$, $n = 20$ neurons) and a significant decrease in the delay of the subsequent spikes (Fig. 4d left, $p = 0.0028$, $n = 20$ neurons). However, there was no effect in L4 neurons in vS1 (Fig. 4c, d, right), consistent with the low connection strength to L4 neurons.

This temporal sharpening effect in pyramidal cells in L2/3 and L5 was consistent with the connectivity profile of the feedback circuits, where feedback projections in both pathways connected to both excitatory neurons and inhibitory interneurons (Fig. 3). Direct connection from feedback to pyramidal cells potentially increases the firing probability right after the onset of the feedback activation, while disynaptic inhibition (PV+ cells and SOM+ cells in V1, and PV+ cells in

vS1) likely mediates the delay in subsequent spiking. In summary, for both LM to V1 and vM1 to vS1, feedback activation temporally sharpened sustained feed-forward excitation by eliciting a transient increase followed by a prolonged decrease in the firing rate of pyramidal cells in L2/3 and L5. This "temporal sharpening" effect is not a unique feature for feedback circuits, but has also been described in feed-forward and local recurrent circuits[2,48–52].

## Feedback facilitates bursting of intrinsically bursty cells when paired with brief feed-forward input in vM1 to vS1 pathway

We next examined the modulatory effect of the feedback activations when paired with brief, temporally precise feed-forward input. Since previous studies have shown that feedback projections synapse on apical dendrites[53] and the activity on apical dendrites facilitates bursting in L5 intrinsically bursty (IB) neurons in S1[54], we hypothesized that feedback regulates the burstiness of L5 IB neurons.

We identified L5 IB neurons in both V1 and vS1 based on their relatively large size and characteristic bursty firing pattern (Supplementary Fig. S3), which corresponded to the thick-tufted cells reported in the literature[55,56]. In a subset of the experiments, we further confirmed the cell identities with their morphologies (Supplementary Fig. S3). To mimic the feed-forward input, we injected a 2 ms current impulse (1.5 nA to 2 nA) to elicit spikes (Fig. 5a, b). On some trials, paired with the feed-forward input, we delivered a 2 ms LED stimulation to activate feedback terminals. We chose an initial delay of 3 ms between the somatic current injection and subsequent optogenetic stimulation in order to maximize the effect on bursting based on

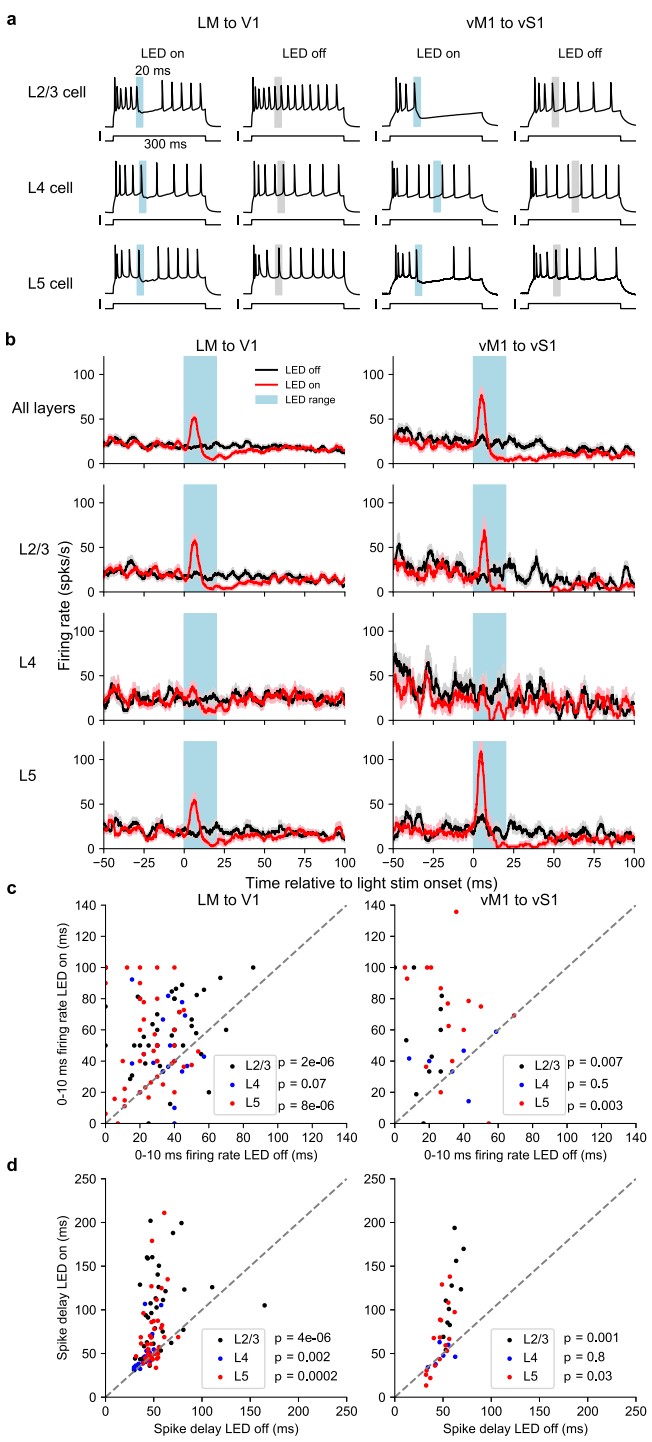

**Fig. 4 | Feedback activity temporally sharpened the firing patterns of pyramidal cells in V1 or vS1 in both feedback pathways. a** Examples of feedback modulation on the firing patterns of pyramidal cells in L2/3 (top), L4 (middle), and L5 (bottom) of V1 (left) or vS1 (right). Cells were driven to fire with sustained positive current injection. In LED-on trials (left plots), 20 ms LED stimulus (blue bar) was delivered. LED-off trials (right plots) were paired with the LED-on trials. Gray bars mark the LED stimulus range of the corresponding LED-on trials. **b** Peristimulus time histogram of pyramidal cells for feedback on trials (red, solid line: mean, shade: s.e.m across cells, same for later) and feedback off trials (black). **c** Firing rate in feedback off trials vs feedback on trials within the time range of 0 to 10 ms after the LED onset for pyramidal cells in L2/3 (black dots, 44 cells over 15 animals for V1 and 11 cells over 2 animals for vS1), L4 (blue dots, 20 cells over 7 animals for V1 and 6 cells over 2 animals for vS1) and L5 (red dots, 45 cells over 15 animals for V1 and 17 cells over 6 animals for vS1). *p* values are from the two-sided Wilcoxon sign-rank tests. **d** Time delay relative to the LED onset of the first spike after LED offset (i.e. after 20 ms) in feedback-on trials vs feedback-off trials for pyramidal cells in L2/3 (black dots, 44 cells over 15 animals for V1 and 11 cells over 2 animals for vS1), L4 (blue dots, 20 cells over 7 animals for V1 and 6 cells over 2 animals for vS1) and L5 (red dots, 45 cells over 15 animals for V1 and 17 cells over 6 animals for vS1). *p* values are from the two-sided Wilcoxon sign-rank tests. Source data are provided as a Source Data file.

forward-only condition (Fig. 5a–c; second spike with feed-forward only: 0.15 ± 0.07, *p* = 0.003, third spike with feed-forward only: 0.09 ± 0.04, *p* = 0.04, two-sided Wilcoxon signed-rank test, *n* = 13 neurons).

According to a previous study[54], the burstiness of L5 IB neurons is highly dependent on the time difference between the stimulation of the cell body (feed-forward) and the apical dendrites (feedback). We therefore varied the time delay (*Δt*) of the feedback onset relative to the feed-forward onset and characterized the corresponding probability of feedback-evoked additional spikes contributing to bursting. Feedback from LM to V1 elicited an extra spike when the feedback was activated after the feed-forward input (Fig. 5a–c left). Other than the *Δt* = 3 ms condition (Fig. 5a, b left), the probability of second spikes at *Δt* = 6 ms (0.54 ± 0.14) was also significantly higher than the probability in the feed-forward-only condition (Fig. 5c, left, *p* = 0.042, two-sided Wilcoxon signed-rank test, *n* = 10 neurons). Importantly, feedback from LM to V1 never elicited a third spike (Fig. 5c, left). However, feedback from vM1 to vS1 increased the probabilities of both a second and a third spike when *Δt* was between 0 and 6 ms (Fig. 5c, right). Regardless of *Δt*, vM1 to vS1 feedback never reduced the probability for the second and the third spike, compared to the feed-forward-only condition (Fig. 5c, right).

Moreover, we also found that for the LM to V1 pathway, the timing of the second spike elicited by feedback was strongly locked to the feedback stimulus onset (Fig. 5d), with a mean time delay (across trials) of 4.4 ± 0.2 ms (mean ± s.e.m across neurons, *n* = 6 neurons, Fig. 5e), and a standard deviation (across trials) of 0.49 ± 0.07 ms (mean ± s.e.m across neurons, *n* = 6 neurons, Fig. 5f). In contrast, the time of the second spike in vM1 to vS1 pathway was both longer and more variable across trials, with a mean time delay of 16 ± 4 ms (mean ± s.e.m across neurons, *p* = 0.0006 compared to LM to V1 pathway, two-sided Wilcoxon rank sum test, Fig. 5e) and a standard deviation of 8.0 ± 2.2 ms (mean ± s.e.m across neurons, *p* = 0.0006 compared to LM to V1 pathway, Fig. 5f).

To exclude the possibility that what we observed is due to the difference in the intrinsic burstiness of IB neurons in V1 and vS1, we directly compared their burstiness by recording the firing patterns of the neurons with step-wise current injections. We measured how prone they were to burst with two metrics: minimal current to elicit a burst and burstiness. Burstiness was defined as the firing rate difference of the spikes inside bursts from that out of the burst, normalized by their summation (see Methods). We found that there were no significant differences between V1 bursty neurons and vS1 bursty neurons in both metrics (Supplementary Fig. S11), indicating that the difference

previous studies[54]. With feed-forward input only, there were low probabilities of bursting in both V1 and vS1 (Fig. 5b, left). With both feed-forward and feedback inputs, we found a major difference in the response of neurons in V1 and vS1. For neurons in V1, LM feedback reliably elicited a single extra spike within 3 ms after LED stimulus onset, with a higher probability of second spikes (0.53 ± 0.15, mean ± s.e.m, same for the rest of the section) compared to the feed-forward-only condition (Fig. 5a–c left, 0.20 ± 0.11, *p* = 0.045, two-sided Wilcoxon signed-rank test, *n* = 10 neurons). Subsequent spiking within at least a 20 ms time window was eliminated. This effect was consistent with the temporal sharpening effects we observed during sustained feed-forward excitation. In contrast, feedback from vM1 elicited bursts of spikes in L5 vS1 IB neurons, with a higher probability of a second (0.60 ± 0.10) and a third spike (0.26 ± 0.06) compared to the feed-

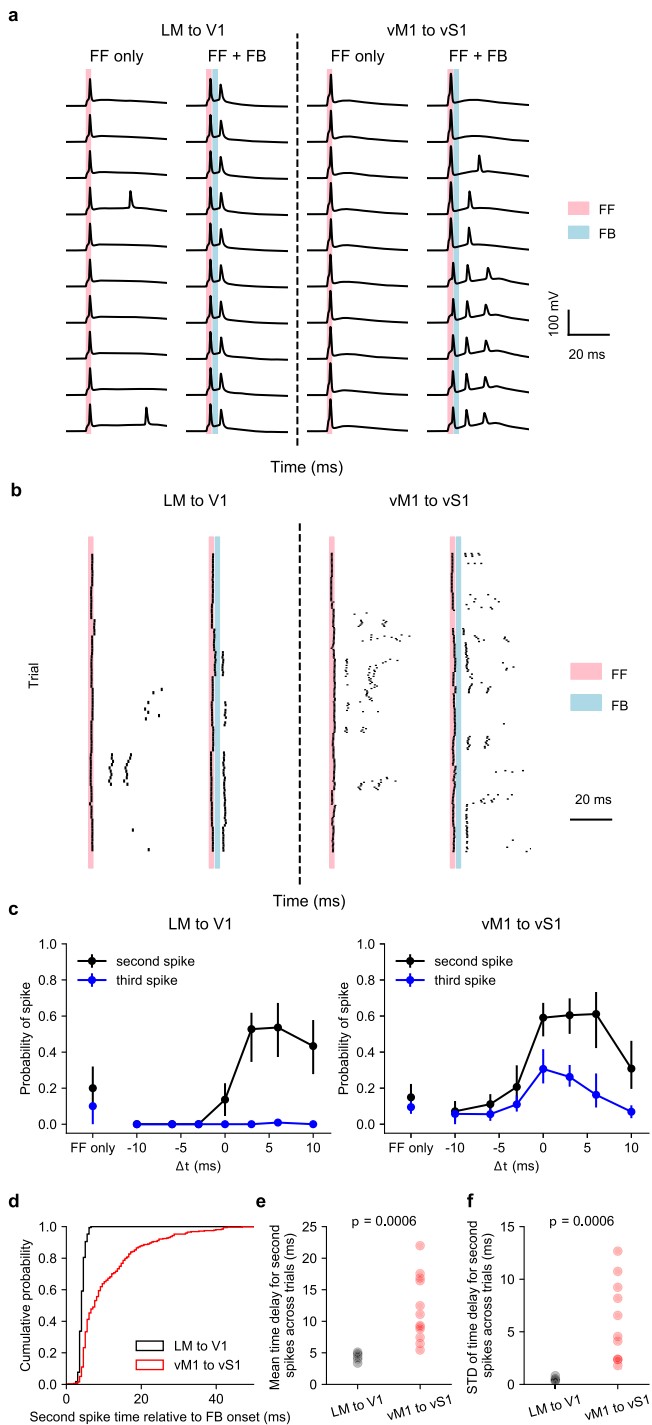

**Fig. 5 | Feedback regulation on bursting behavior of L5 intrinsically bursty (IB) neurons. a** Firing of example L5 IB neurons in response to feed-forward stimulus only (FF, 2 ms) and the combination of feed-forward and feedback stimulus (FF + FB, FB 2 ms) in V1 (left) or vS1 (right). In these examples, the FB stimulus was delivered 3 ms after the FF stimulus. **b** Raster plots of all cells in V1 (left) or vS1 (right). Each row refers to a trial and each tick indicates a spike. **c** Probability of occurrence of the second spike (black) and third spike (blue) of L5 IB neurons responsive to FF + FB stimulus, as a function of the time of FB onset relative to the FF onset (Δt). Dots and error bars are mean and bootstrapped 68% confidence intervals of the mean. $n = 11$ cells over 2 independent experiments for LM to V1 feedback; $n = 15$ cells over 5 independent experiments for vM1 to vS1 feedback. **d** Cumulative probability of second spike occurrence time relative to the FB onset. Black: LM to V1. Red: vM1 to vS1. **e, f** Mean (**e**) and standard deviation (STD, **f**) of the time delay across trials for the second spikes were both significantly higher in the vM1 to vS1 pathway ($n = 6$ cells over 2 independent experiments for LM to V1 feedback; $n = 11$ cells over 5 independent experiments for vM1 to vS1 feedback; $p = 0.0006$, two-sided Wilcoxon rank-sum test). Each dot represents a cell. Source data are provided as a Source Data file.

is transient (2 ms), even though the feedback connection strength is generally weaker for vM1 to vS1 pathway (Supplementary Fig. S7).

## Discussion

In this study, we characterized LM to V1 feedback connections and compared their properties with connections principles described in other studies[23,26]. In contrast with previous reports describing the effects of vM1 to vS1 feedback[23], we observed that LM to V1 feedback targeted L2/3 SOM+ cells instead of VIP+ cells. We replicated the previous results in the vM1 to vS1 feedback pathway as a control to confirm that the differences we observed in feedback in the visual system were not due to our experimental protocol. We also found substantial differences in the layer specificity and the functional impact of feedback connections between these two pathways, arguing against a single universal principle governing the organization and functional operation of feedback in the neocortex. More studies are needed to explore the characteristic connectivity of feedback in other sensory modalities and hierarchies of connected areas, ideally with more high-throughput methods such as electron microscopy or mesoscale connectome techniques[57].

Our results showed that while vM1 to vS1 feedback mainly projected to L1, L5, and L6, LM to V1 feedback innervated all the layers, including some sparse innervations in L4. The observation of projections from LM to V1 L4, although sparse, is in conflict with the canonical notion that feedback terminals avoid L4 in other species such as primates[58] and rats[29,59]. Interestingly, this observation is aligned with a recent study showing that the feedback projection from secondary somatosensory cortex (S2) to S1 also weakly targets L4 in S1[60]. However, further studies are needed to examine whether feedback projections to L4 are a common feature across all feedback pathways within a sensory hierarchy or are unique to feedback from secondary sensory areas to primary sensory areas.

In addition to the connectivity difference in L4, LM to V1 feedback preferentially connected to neurons in L2/3 over L5 while vM1 to vS1 was the opposite, with the exception of SOM+ cells. LM to V1 feedback formed stronger connections to L2/3 SOM+ cells than L5 SOM+ cells, while vM1 to vS1 was the opposite. However, we did not investigate the feedback connectivity to L6 cells in this study. A recent study revealed that among the excitatory cell types in L6, LM to V1 feedback sends strongest projections to the intratelencephalic (IT) neurons that also project to LM while avoiding other excitatory neuronal types, suggesting this may be a defining motif of cortico-cortical feedback in L6[61]. Additional work is necessary to further investigate this topic. Other factors we did not investigate in the current study is the layer specificity of the origin of feedback projections from LM or vM1 and how V1 or vS1 cells respond to activation of feedback terminals residing in different layers. It is important for future studies to dissect the full

we found in bursting with feedback stimulation is not a result of the difference in their intrinsic bursting properties. We also compared the kinetics of light evoked EPSCs of L5 pyramidal cells in V1 and vS1, including the rise time, decay time and time to peak. We found no difference between cells of V1 and vS1 in any of these metrics (Supplementary Fig. S13), suggesting that the difference we found in bursting is not a result of excitatory responses to feedback per se but is related to feedback modulation on the inhibitory circuit.

These results together suggest a major difference in the regulation of L5 IB cell bursting by the two feedback pathways: LM to V1 feedback induces a single, temporally precise extra spike, similar as the pattern of light-evoked EPSP (Fig. 2c) and the temporal sharpening effect with sustained feed-forward input (Fig. 4a, b), while vM1 to vS1 feedback facilitates bursting of L5 IB neurons when feed-forward input

layer-specific connectivity matrix by both specifically expressing ChR2 of LM or vM1 cells in different layers using layer specific Cre lines[62] and specifically activating feedback terminals in different layers using subcellular ChR2-assisted circuit mapping (sCRACM)[45].

Another important difference in the wiring logic between these pathways was that vM1 to vS1 feedback preferentially targeted VIP+ cells over SOM+ cells but LM to V1 was the opposite. Our results are consistent with previous reports describing a disinhibitory feedback circuit from vM1 to vS1[23,50], where feedback targets VIP+ cells that activate SOM+ cells and disinhibit local pyramidal cells[21]. The same circuit has also been found in the feedback from the cingulate cortex to V1[26]. Moreover, a recent study shows that SOM+ neurons in vS1 receive relatively weak long-range connections from S2 as well[50]. In contrast, we found that this disinhibitory circuit was not preserved in the LM to V1 pathway. Instead, feedback from LM to V1 exhibited the opposite pattern, with stronger direct feedback inputs on SOM+ cells than VIP+ cells. This, to our knowledge, is the first evidence that shows SOM+ cells can be strongly activated by top-down feedback projections. Previous studies have revealed an important role of SOM+ cells in surround suppression. Specifically, the activity of SOM+ cells increases with the stimulus size and both activation and inhibition of their activity affects surround suppression in pyramidal cells[63,64]. It has also been shown that higher visual areas contribute to surround suppression in both primates[65,66] and rodents[67,68]. In light of these previous findings, our results suggest that the effects of feedback on surround suppression may be mediated by direct feedback connections on SOM+ neurons in V1.

Functionally, we found that when combined with sustained feed-forward input, both feedback pathways "temporally sharpened" feed-forward excitation by eliciting a transient increase followed by a prolonged decrease in the firing rate of pyramidal cells. This "temporal sharpening" was consistent with previous studies in a variety of contexts[47,49,69–74], suggesting that it is a canonical circuit motif. The mechanism of the inhibitory phase of the biphasic response could include direct[48] or indirect[75] excitation of local interneurons from the long-range projections. In our case, given the connectivity we described here, the most likely explanation of the inhibitory phase we observed is the direct excitation of interneurons from LM or vM1 feedback projections and not the indirect excitation of inhibitory cells via local V1 or vS1 excitatory neurons. Despite simultaneously driving a large number of feedback terminals with optogenetics, feedback excitation never elicited spiking in V1 or vS1 excitatory neurons, and thus did not excite interneurons via local excitatory neurons.

Compared to the sustained feed-forward inputs, transient feed-forward inputs may better mimic rapidly changing feedforward inputs that occur under ethological conditions. Here, we showed that when paired with transient temporally precise feed-forward input to the cell bodies, only feedback from vM1 to vS1 increased the probability of bursting in L5 IB cells, while feedback from LM to V1 continued to show temporal sharpening. We characterized the difference in bursting between the two feedback pathways by two metrics: (a) the probability of eliciting a third spike and, (b) the latency distribution for the second spike relative to the LED onset. We saw significant differences in both of these metrics. In the LM to V1 pathway, feedback typically elicited a single reliable spike that was time-locked to the optogenetic stimulation. This pattern is consistent with the light-evoked EPSP recording of L5 neurons (Fig. 2c) and temporal sharpening experiments (Fig. 4a), where there was a sharp window of excitation followed by a prolonged inhibitory phase. In contrast, in the vM1 to vS1 pathway, feedback triggered additional spikes that fired at more random times relative to the optogenetic stimulation onset. This effect was very different from the spikes elicited in the LM to V1 pathway where spikes were locked to the optogenetic stimulus (Fig. 5b, d–f). The difference in the connections to SOM+ cells and VIP+ cells between the two feedback pathways may play an important role in establishing these different firing patterns

in the two areas. VIP+ cells inhibit SOM+ cells that have been shown to target and gate the activity on the apical dendrites[76–78]. Therefore, without the inhibition from SOM+ in the vM1 to vS1 feedback pathway, the input on the apical dendrites from the feedback projections may induce plateau potentials, propagated to the soma to elicit bursting[54,55]. Previous work revealed that feedback from vM1 to vS1 elicits calcium spikes at the apical dendrites[79], and simultaneous activation of the apical and somatic compartments elicit bursting in L5 IB neurons[54]. Here we linked these two findings together by directly showing that the combination of feed-forward and vM1 to vS1 feedback inputs were able to elicit bursting in L5 IB neurons in vS1. Given that bursts activate their targets stronger than single spikes, which effectively depolarize their targets to be more sensitive to detect subsequent inputs[80,81], vM1 to vS1 feedback may also function as an attentional signal that amplifies the sensory responses. In the LM to V1 feedback pathway, in contrast, the strong connections to SOM+ cells might explain why we did not observe bursting facilitation on V1 L5 IB cells. Plateau potentials and calcium spikes on the apical dendrites could be inhibited by SOM+ activities[82], preventing bursts in the cell bodies. However, the causal relationship between feedback-activated disinhibitory circuit and bursting requires further investigation with optogenetic inhibition of SOM+ cells or VIP+ cells while activating feedback. In addition to SOM+ cells, we also found that feedback from LM to V1 targeted PV+ cells, consistent with previous reports[25,83–85]. In contrast to SOM+ cells, the majority of PV+ cells target the soma and basal dendrites of pyramidal cells. The fact that feedback from LM to V1 connected to both PV+ cells and SOM+ cells suggests that this feedback precisely controls the time of activities not only at the cell body but also at the apical and tuft dendrites. Given the precise time window of the excitation, feedback from LM can only enhance the activity in V1 in cases where there is a temporal coincidence between higher-level representations in LM and sensory information in V1.

Our work leads to a question about what information V1 neurons integrate from LM feedback that requires such precise timing. One of the leading models that have been proposed for the feedback integration is the predictive coding model[4,5,86]. In this model, feedback is interpreted as the prediction signal while the feed-forward input is the prediction error[4]. With this notion, Friston[87] predicted two different types of feedback projections. One type is to deep layers which is suppressive, and the other is to superficial layers which is modulatory and excites similar features but suppresses orthogonal features. We found that LM to V1 feedback projected to both pyramidal cells and interneurons (i. e. SOM+ cells and PV+ cells) in L2/3 which could serve as a modulatory signal with both excitatory and inhibitory components. In contrast, L5 pyramidal cells mostly received inhibition from feedback, thus being more suppressive, which is consistent with the predictions of the predictive coding model. However, in our study we activated all feedback terminals simultaneously which did not likely preserve the natural pattern of feedback activity. Therefore, future in vivo work is needed to examine how LM neurons precisely influence V1 neural activity under natural sensory stimulation and behavioral conditions, and how this process is regulated by interneurons. Several recent studies have explored this direction. In particular, Marques et al., 2018 shows that compared to local V1 neurons, LM to V1 feedback boutons over-represent the visual areas perpendicular to their preferred orientation of the boutons, but are aligned with their preferred direction[39]. Building on the results we show here, it will be intriguing to further dissect the microcircuitry to directly test the predictions for the predictive coding model.

## Methods

### Animals and surgeries

All procedures performed on animals were in accordance with the ethical guidelines of the National Institutes of Health and were

approved by the Institutional Animal Care and Use Committee (IACUC) of Baylor College of Medicine, with the animal protocol number AN-4703.

In this study, we used 79 mice in total (51 males and 28 females), aged 8 weeks to 4 months. These included 4 C57Bl/6 mice (all male), 29 PV-Cre/Ai9 mice (19 males and 10 females), 35 SOM-Cre/Ai9 mice (22 males and 13 females), and 11 VIP-Cre/Ai9 mice (6 males and 5 females). All animals were maintained in the animal facility with a light-cycle from 6 am to 6 pm daily, temperature ranging from 68 to 72 °F and humidity ranging from 30% to 70%. All Cre and Ai9 reporter lines are on a C57Bl/6 background, and they are from Jackson Labs as follows:

SOM-Cre: https://www.jax.org/strain/013044
VIP-Cre: https://www.jax.org/strain/010908
PV-Cre: https://www.jax.org/strain/008069
Ai9 reporter: https://www.jax.org/strain/007909

Before each experiment, we performed the following surgical procedures on the animals. We used 3% isoflurane to induce anesthesia, and anesthetized animals were placed in a stereotaxic head holder (Kopf Instrument). The anesthesia was then maintained with 1.5–2% isoflurane and the body temperature was maintained at 37 °C during the whole surgical procedure using a homeothermic blanket system (Harvard Instrument). We injected the following drugs at the beginning of the surgery: 0.05 mL, 0.5% bupivacaine subcutaneously under the scalp, 3 mg/kg dexamethasone intramuscularly in the leg, and 7.5 mg/kg ketoprofen subcutaneously on the back. After 10–20 min, we removed an approximately 1 cm² area of skin above the skull and cleaned up the underlying fascia. With the surgical glue (VetBond, 3 M), we sealed the wound margins. We then attached a custom-made head bar on the skull with dental cement (Dentsply Grip Cement). After the dental cement was completely dry, we removed the mouse from the stereotaxic frame and held the skull stationary on a small platform with the newly attached head bar. Using a surgical drill and HP 1/2 burr, we made a ~3 mm diameter craniotomy on the right hemisphere with a center 3 mm lateral of the midline and contacting the lambda suture on its posterior edge, which allowed the exposure of areas V1 and LM. The exposed cortex was then cleaned up with ACSF (125 mM NaCl, 5 mM KCl, 10 mM Glucose, 10 mM HEPES, 2 mM CaCl₂, 2 mM MgSO₄). After viral injections (described in the section Virus Injections), the cortical window was sealed with a 3 mm-diameter coverslip (Warner Instruments), using VetBond glue.

### Intrinsic optical imaging and visual areas identification

We used intrinsic imaging to identify the precise locations of V1 and LM (Fig. 1b). The animal was kept anesthetized with 1–2% isoflurane. We measured the change in cortex reflectance to red light with a wavelength of 610 nm[37]. Using a CCD camera, we captured 512×512 pixels images at a rate of 12 Hz. To present visual stimuli, we positioned a 7" LCD monitor (Lilliput 665GL-70NP/HO/Y monitor, 60 Hz scan rate) approximately 10 cm away from the left eye of the animal, covering about 88° (azimuth) and 72° (elevation) of the contralateral visual field. To map the retinotopy of V1 and LM, we stimulated the animal with drifting gratings presented in one of the four locations (top lateral: −20° azimuth, 20° elevation; top medial: 20° azimuth, 20° elevation; bottom lateral: −20° azimuth, −20° elevation; bottom medial: 20° azimuth, 20° elevation) on each trial. The gratings were drifting either vertically or horizontally, with a spatial frequency of 0.03 cycles/°, a temporal frequency of 4 Hz, and a size of 8°. Stimuli were presented for 2 s and separated with a 3-second luminance-matched gray background. Stimulus displays were generated with MATLAB Psychophysics Toolbox and a photodiode attached to the screen that allowed a precise time-stamping of each frame of stimulus presentation on the clock of CCD camera recording. We then constructed the retinotopic map based on the brain reflectance using linear regression and identified V1, LM, and AL by comparing with the published retinotopic maps[33].

### Virus injections and histology

We used adeno-associated virus (AAV, serotype 2/1; University of Pennsylvania Gene Therapy Program Vector Core, titer 2.3 × 10¹² /mL) to express ChR2-EYFP under the CaMKIIα promoter, which only allows ChR2-EYFP to express in the excitatory neurons in LM or vM1.

Anatomical structures used in this study (V1, LM, vS1, vM1), are shown in Fig. 1a. To inject in LM, we made the injection in one of the four retinotopic locations of LM, with the guidance of the vessels and the retinotopic map obtained with intrinsic imaging. For vM1, we stereotactically injected with coordinates 1.1mm anterior and 0.9 mm lateral relative to bregma (Fig. 1a). Virus was injected into two depths in the cortex, 300 μm and 700 μm below pia, with a volume of 30 nL each, aiming to cover cells in all layers.

Two to four weeks after the injections, we checked the virus infection by imaging the EYFP expression pattern under a two-photon microscope (Fig. 1c). We took two-photon 300 μm stacks, covering the whole craniotomy.

In some experiments (3 animals for LM injections and 4 animals for vM1 injections), we checked the laminar distribution of axon terminals from feedback projections. Before the perfusions, we injected 2% Fast Green FCF (Sigma-Aldrich) stereotactically into the area in V1 or vS1 that had the densest expression of ChR2-EYFP. Fast Green FCF is blue under the widefield imaging and does not have any fluorescence, which serves as a marker to guide our slicing and imaging on the slices. We then perfused the animal with 4% paraformaldehyde (PFA) via the cardiovascular system. We removed the brain from the skull and left it in the 4% PFA solutions overnight. On the second day, the brain was cut into 50 μm coronal slices using vibratome (Leica, VT1200S). In some experiments, to facilitate the identification of different layers, we used DAPI containing mounting media to mount the brain slices and stained the nuclei (VectaShield, H-1200-10). The regions of interest on the slices (injection sites in LM or vM1 or the projection sites in V1 or vS1) were imaged under either two-photon microscopy or epifluorescent microscopy.

### Multi-cell whole patch-clamp recording in brain slices with optogenetics

We followed the protocol described in previous studies[14] to prepare the visual cortical slices. NMDG (N-Methyl-D-glucamine) was used in the slicing solutions to improve the quality of slices from adult animals[43]. Before the experiments, similar to the procedures before histology, we also injected 2% Fast Green FCF stereotactically into the area in V1 or vS1 with the densest expression of ChR2-EYFP. Then the animal was put into deep anesthesia with 3% isoflurane and decapitated. The brain was quickly removed and placed into 0–4 °C oxygenated NMDG solution (93 mM NMDG, 93 mM HCl, 2.5 mM KCl, 1.2 mM NaH₂PO₄, 30 mM NaHCO₃, 20 mM HEPES, 25 mM glucose, 5 mM sodium ascorbate, 2 mM Thiourea, 3 mM sodium pyruvate, 10 mM MgSO₄ and 0.5 mM CaCl₂, pH 7.35). We cut 300 μm thick parasagittal slices from the tissue blocks with a microslicer (Leica VT 1200). With the guidance of the injected dye, we only kept the slices that contain our region of interest in V1 or vS1 (LM or vM1 was not in the slices), which was marked with Fast Green. Slices were kept at 37.0 ± 0.5 °C in oxygenated NMDG solution for 10–15 min and then transferred to the normal ACSF (125 mM NaCl, 2.5 mM KCl, 1.25 nM NaH₂PO₄, 25 mM NaHCO₃, 1 mM MgCl₂, 25 mM glucose and 2 mM CaCl₂, pH 7.4) for 0.5–1 h before recording. During the recording, the slices were submerged in a chamber and stabilized with a fine nylon net attached to a platinum ring. The recording chamber was filled with oxygenated ACSF.

We performed multi-cell whole-cell patch recordings in L1, L2/3, L4 and L5 in the region of interest in V1 or vS1, with 4–7 MΩ borosilicate pipettes (2.0 mm OD, 1.16 mm ID, Sutter Instruments) filled with a standard low-chloride internal solution (120 mM potassium gluconate, 10 mM HEPES, 4 mM KCl, 4 mM MgATP, 0.3 mM Na₃GTP, 10 mM sodium phosphocreatine and 0.5% biocytin, pH 7.25). For IPSC

recordings, a Cs+ based internal solution was used instead (135 mM cesium methanesulfonate, 10 mM HEPES, 2.5 mM MgCl$_2$, 4 mM Na$_2$ATP, 0.4 mM Na$_3$GTP, 10 mM sodium phosphocreatine, 0.5 mM EGTA, 0.1 mM spermine and 0.5% biocytin, pH 7.25). On each slice, we always recorded at least one L2/3 pyramidal cell as an internal reference. For voltage-clamp recordings, we clamped the voltage at −85 mV to record EPSCs. For current-clamp recordings, we adjusted the membrane voltage to −70 mV (Fig. 2b–e). We performed both EPSC and EPSP recordings because they lead to different interpretations. EPSCs characterize the strength of synaptic connection from feedback terminals to cells in V1 or vS1, while the EPSPs reflect a final result of activities of a neuron evoked by the feedback stimulation, including the effect of both the monosynaptic excitation and disynaptic inhibition. With both measures, we were able to estimate both relative strengths of connections and the network effect of the feedback activities.

To activate ChR2 expressed in the axon terminals, we delivered 470 nm LED blue light through the light path of the microscope with a 40x objective lens. The LED light spot with a spatial spread of approximately 1 mm$^2$ covered all layers of the cortex in the recorded slice. 2 ms LED pulses with an intensity of approximately 6 mW/mm$^2$ were used to trigger EPSCs or EPSPs (Figs. 2 and 3).

In some experiments to identify monosynaptic events (Supplementary Fig. S5), we applied TTX (1 μM) to block action potentials, and then applied 4-AP (0.5 mM) to block potassium channels to enhance the responsiveness of ChR2 expressing axon terminals. If the EPSC was recovered with 4-AP, we regarded it as a monosynaptic event.

Cell classes were primarily identified by their genetic markers and were further confirmed with electrophysiological and morphological features (Supplementary Figs. S2 and S3). Cell morphologies were reconstructed following methods reported in previous studies[14,76,88]. Briefly, after the electrophysiological recording, slices were immersed in freshly prepared 2.5% glutaraldehyde/4% paraformaldehyde in 0.1 M phosphate-buffered saline at 4 °C for at least 48 hours and were subsequently processed with the avidin-biotin-peroxidase method to recover the morphologies. The morphologically recovered cells were examined, reconstructed and analyzed using a 100X oil-immersion objective lens and a camera lucida system. Multiple whole-cell recording allows us to recover the morphologies of multiple cells recorded in one slice simultaneously, which is essential to our study because it gives us another dimension to identify neuronal types, especially in the cases where genetic markers are not precise enough to classify neurons with different functional features. Several recent studies have shown that SOM-Cre lines also label some fast-spiking (FS) cells[40,89,90]. FS cells were basket cells, which were very different in both morphology and firing patterns from other SOM + types[91] (Supplementary Fig. S2). We, therefore, excluded FS cells in the SOM + group in later analyses.

In experiments for temporal sharpening (Fig. 4), we injected 200−800 pA positive current lasting 300 ms to drive the cells to fire with a regular pattern. On LED-on trials, we delivered 20 ms 470 nm LED stimuli in a random time, ranging from 100 ms to 200 ms after the onset of the current injection. LED-off trials were interleaved with the LED-on trials, to control for the effect by the firing pattern change across time.

In experiments for bursting (Fig. 5), we identified the L5 intrinsic bursty neurons by their laminar position, large cell bodies, and thick apical dendrites. After patching the cells, we then preselected IB neurons by their firing patterns (Supplementary Fig. S3) with stepwise current injections of 700 ms, from −200 pA to 600 pA, with 20 pA steps in between. We then injected 2 ms 2000 pA positive current to elicit a single spike. The intrinsic bursty neurons sometimes fired a burst of spikes in response to the current injection only. The strength of the feedback activities varied across animals and slices because of the variance in the amount of virus injected. To make the results comparable across experiments, we recorded the light-evoked EPSCs before the bursting protocol and adjusted the LED light intensity to

elicit EPSC ranging from 100 pA to 500 pA. Light-evoked EPSCs between the intrinsic bursty neurons in V1 and vS1 have no significant difference (Supplementary Fig. S12). On LED-on trials, we delivered 2 ms 470 nm photostimuli with various time differences relative to the current injection: −10, −6, −3, 0, 3, 6, 10 ms. For each time difference, we recorded 6 to 10 trials.

### Data analysis for EPSC or EPSP recordings

**Amplitude and latency measurement.** The baseline activity was defined as the mean activity within 40 ms prior to the LED stimulus onset. The peak value of EPSC or EPSP was the maximum value within 35 ms after the LED stimulus onset relative to the baseline activity. If the cell fired in response to the light stimulus, we then defined the EPSP amplitude as the difference between the baseline membrane potential and threshold potential. The latency of the synaptic event was estimated from the extrapolated intersection of the baseline with a line through the two points of time when the current was 20% and 80% of the peak value[92,93].

**Criteria of cell "responsiveness".** We considered a cell responsive to light stimulation (Fig. 3a) when both criteria were met: first, the latency of the response (EPSC or EPSP) is less than 7 ms; second, the amplitude of the response is larger than 3 times the standard deviation of the baseline activity. We selected 7 ms as a cutoff based on our experiments with TTX and 4-AP, there are no events with a latency greater than 7 ms that were recovered with the addition of 4-AP and rarely any events smaller than 7 ms that were not recovered with the addition of 4-AP (Supplementary Fig. S5c).

**Statistical comparison of EPSC or EPSP values.** To compare the amplitudes of EPSCs and EPSPs across different slices and animals, we normalized the amplitude of individual cells to the amplitude of the L2/3 pyramidal cell on the same slice. If there were more than one L2/3 pyramidal cells recorded on the same slice, we normalized by their mean amplitude.

We statistically compare the log2 normalized amplitudes of EPSC or EPSP between different cell classes across slices within the same feedback pathway. We performed Conover's test[94] for comparisons among cell classes other than L2/3 pyramidal cells, which is the *post hoc* pairwise comparison followed by Kruskal−Wallis one-way analysis of variance by ranks. We performed a permutation signed test for comparisons between L2/3 pyramidal cells and other types.

To statistically compare the normalized amplitudes of EPSC or EPSP of the same cell type across different pathways, we performed the Wilcoxon rank-sum test, with *p* values corrected by Benjamini-Hochberg adjustments for multiple comparisons.

**Criteria for "spiking cells".** To quantify the probability of neurons to fire in response to light stimulation (Fig. 3c), we only included data from slices with a minimal level of ChR2-expressing feedback innervation, defined by a minimum mean EPSC in L2/3 pyramidal cells of 50 pA. We never observed spikes on slices that did not meet this minimum criterion.

### Data analysis of experiments with a sustained current step

For the experiments with sustained current steps (Fig. 4), we detected the spikes by thresholding and aligned the spike trains with the onset of LED pulses. We computed mean firing rates across trials for each cell, which were smoothed by convolving with a box-car filter of 4 ms time bins. To quantify the excitatory effect of the feedback modulation, we compared the firing rate of LED-on trials and LED-off trials within 10 ms after the LED stimulus onset of the LED-on trials or the matching time points of the LED off trials. To quantify the inhibitory effect of the modulation, we compared the spike delay of the first spike after the LED-on period of the LED-on trials or the matching LED-on period of

the LED-off trials, relative to the LED stimulus onset. We performed the Wilcoxon signed-rank test for the statistical comparison.

## Data analysis of bursting experiments

**Characterize the intrinsic burstiness of neurons.** Given strong differences in dendritic integration properties of L5 cells that has been previously reported[95], it is possible that L5 neurons are more prone to burst in vS1 than in V1. To exclude the difference in the intrinsic burstiness, we preselected IB neurons by their firing patterns with stepwise current injections of 700 ms, from −200 pA to 600 pA, with 20 pA steps in between. To characterize their intrinsic burstiness, we defined two metrics: minimal current that elicits bursts and burstiness. Following the method described in Allen Software Development Kit (github.com/alleninstitute/allensdk), burstiness was defined as the firing rate difference of the spikes inside bursts from that out of the burst, normalized by their summation. We took the median of this normalized firing rate difference over the traces of the first 5 current steps that elicited bursting and used this as a "burstiness" metric. We also quantified the minimal current that elicits bursts for each cell, which indicates how prone a cell is to bursts in response to current injections. We found no difference in either of the two metrics between L5 IB cells in either V1 or vS1 (results in Supplementary Fig. S11).

**Statistical analysis for bursting with feedback activation.** For the bursting experiments (Fig. 5), we performed two analyses. First, we showed the probability of the occurrence of the second spike and third spike as a function of the time difference between the feed-forward and feedback inputs (Fig. 5b). We performed the Wilcoxon signed-rank test to compare the probability of second spikes and third spikes with feedback stimulus delivered at different time points after feed-forward onset to those with feed-forward stimulus only. Second, for those trials where a second spike occurred, we characterized the time delay of the second spike relative to the feedback onset. For each cell that had at least a second spike across trials, we compared both the mean and the standard deviation of the time delay in the two feedback pathways with the Wilcoxon rank-sum test.

## Reporting summary

Further information on research design is available in the Nature Research Reporting Summary linked to this article.

## Data availability

The processed data in this study have been deposited in Figshare 10.6084/m9.figshare.21086572. Access to the full data pipeline of this study is available from the corresponding author on reasonable request. Raw data of all figures are provided in the Supplementary Information/Source Data file with this paper. Source data are provided with this paper.

## Code availability

The code to generate the essential figures are available at GitHub repository https://github.com/shenshan/feedback_paper_codeshare, archived in https://doi.org/10.5281/zenodo.7106045. The code to generate the visual stimuli are available at GitHub repository https://github.com/shenshan/visual_stimuli with https://doi.org/10.5281/zenodo.7106033. We use DataJoint[96] to manage the data collection and data analysis, which is an open source framework available at https://github.com/datajoint/datajoint-python for python version and https://github.com/datajoint/datajoint-matlab for matlab version.

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

## Acknowledgements

We thank Dr. Emmanouli Froudarakis, Dr. Dimitri Yatsenko, Dr. Saumil Patel, Yves Bernaerts, Francisco A. Blanco, and other lab members of Tolias lab for technical support. We thank Dr. Philipp Berens, Stelios Papadopoulos for discussions and comments on the manuscript. This study is supported by the Intelligence Advanced Research Projects Activity (IARPA) via Department of Interior/Interior Business Center (DoI/IBC) contract number D16PC00003. This work was also supported by the National Eye Institute and National Institute of Mental Health under Award Numbers R01EY026927, R01MH109556 and U19MH114830. This research was also supported by NEI/NIH Core Grant for Vision Research P30EY002520.

## Author contributions

S.S., X.J., F.S., and A.S.T. designed the experiments. S.S., X.J., F.S., J.F., P.F., Z.T., and N.Z. performed the experiments. S.S., X.J., D.K., and F.S. analyzed the data. S.S., X.J., F.S., D.K., F.S., J.R., and A.S.T. wrote the manuscript. A.S.T. supervised all aspects of the project.

## Competing interests

The authors declare no competing interests.
