## [Peer Review File · Nature Communications]

Distinct organization of two cortico-cortical feedback pathwaysREVIEWER COMMENTS

Reviewer #1 (Remarks to the Author):

GENERAL COMMENTS

This is a very nice study demonstrating that feedback (FB) circuits are not anatomically and functionally homogeneous across the brain. Using optogenetic stimulation of FB from either M1-to-S1 or LM-to-V1 combined with whole-cell patch recordings in S1 and M1, respectively, in mouse cortex, the authors show that these two FB pathways target different cell types and modulate excitatory neuron activity in distinct ways. The results are novel, important, of broad interest and the study is well executed. I have several suggestions for improvement, but overall find this an excellent study.

MAJOR SPECIFIC COMMENTS

1. Results p. 3 and Fig.1. The authors find distinct feedback (FB) laminar termination patterns in V1 vs. S1. However, based on what we know from primate visual cortex, it is very possible that the LM-to-V1 pattern seen here is unique to FB from V2 to V1, rather than a general rule of FB connectivity within visual cortex. The authors should comment about this in the discussion.
2. Fig. 1B. I suggest to lightly outline the areal borders in the optical maps, as this makes it easier to see how the areas were identified in C.
3. Fig. 1D. The conclusion that M1 to S1 FB connections are less topographically organized than LM-to-V1 seems unwarranted. The comparison should be made for FB terminals within V1 only vs. within S1 only. If the dashed outline in D represents all of S1 then FB to S1 is also somewhat topographically organized, as only a region of S1 shows green label. Also there is no quantitative analysis similar to that done in panel B for the M1-to-S1 projection. This analysis should be done or the claim that the latter FB is less topographic should be dropped.
4. Fig. 1E. Some form of staining that demonstrates layers and area specificity should be added next to each micrograph of the fluorescent label.
5. Was the location of the injection sites confirmed in histological sections to determine whether the results presented here apply to both FB arising from superficial as well as deep layers? Images of at least some injection sites should be shown as supplementary material.
6. Methods P.19. It is stated that whole-cell patch recordings were performed only in L2/3, 4 and 5. But this is inconsistent with several figures demonstrating also L1 inhibitory cells were recorded from. It is a real shame that there are no L6 cell recordings as this is a major target of FB connections.
7. Methods P.19 and Fig.S3. The criteria used to determine monosynaptically evoked EPSCs and EPSPs seem somewhat weak. First it is stated that to identify monosynaptic events TTX and 4-AP were applied to the slices in a "subset of the experiments". Please specify what fraction of experiments this was done for. As a second criterion the authors used 8ms latency of evoked events as the cutoff latency to identify monosynaptic connections. However, in Fig. S1C the appropriate latency would seem to be <4ms, as this is the only latency that does not show overlap between recoverable and non-recoverable cells. The analysis should be repeated using 4 ms as the cutoff latency. Moreover, what is the 3/83 of recoverable events mentioned in the legend? This number does not seem to agree with any of the figures in panels B-C. Finally in panel C what does the Y axis measure? Trace counts? Does this correspond to cell numbers? if not please define it.
8. What is the LED light intensity used for photostimulation (in mW/mm²), and were control experiments performed in the absence of opsins to determine potential light-induced effects?
9. What was the spatial spread of LED activation? Were the authors activating FB across all layers, or were they able to focally activate the FB in different layers? I am wondering here whether the authors could distinguish between FB projecting to different layers. This question is motivated by the possibility that there exist different sets of FB connections related to different layers that could have different impacts on postsynaptic cells. If the authors did not distinguish between effects of FB to different layers, this should be made clear in the discussion, particularly in the context of predictive coding models, which indeed postulate different FB effects in different layers.
10. Results pp 7-8 and Fig.3. There seem to be more interesting laminar differences within the same FB pathway than the authors point out. For example, the effects on SOM or excitatory cells in V1

seem to differ in L2/3 vs L5. Any superficial vs deep layer differences should be emphasized, as there are theories that FB to different layers serve different functions. The data in Fig.3D should also be split by layers for the same reason. In this respect it is a shame that no recordings were performed in L6 as this is a major target of FB connections.

11. Results p. 10 and Fig.4. It would be very interesting (and revealing about underlying mechanisms) to know how the inhibitory neurons responded to these experimental manipulations. Didn't the authors record from inhibitory neurons too during these experiments?

12. Discussion p. 15, 2nd paragraph :” Previous studies have revealed that SOM+ cells are involved in surround suppression because their activities increase with the stimulus size...”. But also because the same cited studies showed that inactivation of SOM cells reduces surround suppression.

13. Discussion p. 15, 2nd paragraph:” Our finding here filled in this gap by showing that feedback from LM to V1 recruits local SOM+ interneurons.”. The authors have not filled this gap, as they have not studied any effects of FB or SOM neuron activity on surround suppression. It would be more accurate here to say something like” our study suggests that the effects of FB on surround suppression are mediated by direct FB contacts with SOM neurons”.

14. Discussion p. 16, 2nd paragraph. It is not correct to state that Predictive Coding models view FB as suppressive. The Predictive Coding model proposed by e.g. Friston and colleagues, call for two different kinds of FB. One FB, to L6, is suppressive and driving, the other, to superficial layers, is modulatory and excites similar features but suppresses orthogonal features.

15. The discussion section does not read very well, and needs to be improved in both English as well as content. The section on predictive coding is not well related to the results of this study.

MINOR SPECIFIC COMMENTS

1. Please number the pages. It makes it easier for Reviewers.

2. Page 2. “Among GABAergic interneurons,.....GABAergic interneurons”. Please specify these facts apply to mouse cortex, as for example it is not known whether these 3 classes of neurons are non-overlapping or whether they represent 80% of GABAergic neurons in primates. Also Ref. 15 is limited to VIP neurons; it would seem more appropriate here to cite a few more comprehensive reviews.

3. Fig. S1 legend. Typo “bursty” not “busty”.

4. Results. p. 8 line 12. The correct figure citation here is 3D not 3C.

5. Results. p.9 bottom of page, paragraph title. The term principal cells here is inappropriate. All that you know is that these are excitatory cells, with no knowledge of whether these

Reviewer #2 (Remarks to the Author):

Shen and colleagues compared two feedback projections in the neocortex of the mouse. The first is a projection from vibrissal motor cortex (vM1) to barrel cortex (vS1), and the second is a projection from the lateral-medial extrastriate area (LM) to the primary visual cortex (V1). They used optogenetics and patch clamp recordings to map the cell type-specific strength of the feedback input. They find that, while vM1 -> vS1 projections preferably contact vasointestinal peptide-positive neurons, the LM -> V1 projections preferably contact somatostatin-positive cells. They went on to determine the functional consequences. They found that coincident activation of feed-forward input to pyramidal cells in S1, and feedback input from vM1, were more likely to evoke a burst response compared to the same stimulation protocol in the visual area.

Altogether, I did not find this work suitable for Nature Communications for the following reasons:

1) Assuming that the results are accurate, I am not sure whether this work presents a sufficient conceptual advance to warrant publication in Nature Communications. First, the logic, or argument, for comparing these two pathways is not clear. It seems to me that one could have picked any two top-down pathways and compared them, and it would not have surprised me if there were always some differences.

It is also not entirely clear why the authors focused on the differences in connection strength between VIP and SOM neurons. They also found many other differences between other cell types, and these may even be more functionally relevant.

2) On the technical side, I am not convinced that the observed differences are not due to the experimental design.

First, the authors elaborated on the issue of polysynaptic activation. While the pyramidal cells do not typically fire when stimulating the feedback projections, the inhibitory cells often do. This may be due to forward inhibition and could affect the amplitude of the excitatory responses in a cell type-specific manner. Examples of this may be seen in Fig.2E; The LM->V1 projections triggers an EPSP that is rapidly followed by an IPSP. This seems not the case for the vM1->S1 projection. While this is, of course, just a single example, it is not clear to what degree this will affect the amplitude of responses and distort the results.

Second, and related to the point just above, the full-field blue light stimulation in figs 4 and 5, may trigger complex interactions between cells in the network. Hence, it is not obvious to relate the functional differences between the feedback projections to the differences in the wiring diagram. Is this really due to the differential input strength to VIP and SOM cells?

Third, the authors do not report the light density used to activate the opsin. In particular, for figs 4 and 5, the light intensity may change the outcome. I would recommend performing a light intensity input-output relationship to see whether conclusions hold for different light intensities.

Fourth, the authors note that the strength of connections from vM1 is generally weaker than the LM projection. The former is, however, a longer projection and opsin expression density at the axon terminals may, therefore, be weaker.

3) It is not clear to me whether the stimulation protocols in Figs.4 and 5 bear any relevance for in vivo activity. We still know very little about these projections. We do not know when these feedback projections are active during behavior, and we do not know whether these projections fire rate-coded output or temporally precise brief feedback. In addition, the disadvantage of using full-field optogenetic stimulation is that all feedback projections are activated simultaneously, which is a sledge-hammer effect for the network, and which most likely does not bear relevance for how the circuit operates.

4) The manuscript is not an easy read. I would recommend moving many of the technical details to the methods section to help the reader focus on the concepts and results. Right now, the message is drowning in the technical details.

Reviewer #3 (Remarks to the Author):

Using in vitro electrophysiology and optogenetics, Shen and colleagues characterize the connectivity profiles of two intra-cortical feedback pathways. The authors show that projections from primary motor cortex (M1) in primary somatosensory cortex (S1) make stronger connections onto vasointestinal peptide (VIP), while feedback projections from the lateromedial visual area (LM) to primary visual cortex (V1) makes stronger connections onto somatostatin (SOM) positive interneurons. Moreover, M1 projections are less connected with layer 4 cells of their target region. For both pathways, activating feedback projections induces a phase of brief spiking followed by a suppression of activity in the target region. When optogenetic stimulation of feed-back projections is combined with a brief current injection in the target neurons (called incorrectly feed-forward input), M1 input onto S1

neurons is more likely to induce a third spike, while LM input onto V1 neurons induces maximally two spikes, but with less jitter than for M1 to S1 connections.

Numerous types of long range cortico-cortical connections have been described but it remains unclear what rules govern their organisation. In depth characterisation of the different circuit motifs that can be recruited by these projections is therefore important to further our understanding of cortical communication. However, I have major concerns about the interpretation of some key results of this study:

Main concerns:

1) The authors claim that the difference they observe is a general distinction between intra-modality feedback and motor to sensory feedback (page 2 and 15). However, they compare different input areas (M1 and LM) targeting different output areas (S1 and V1). From the presented data alone it is therefore impossible to make this claim. Differences observed could be due to differences in circuit and cellular properties of V1 and S1 rather than due to the modality of the feedback projection (see also point 2). Ideally the authors should compare sensory and motor feedback onto the same target area and neurons, e.g. M1 and S2 onto S1. If this is beyond the scope of this study, at least the authors need to tone down their claims and discuss this issue carefully in the discussion.

2) The authors claim that M1 to S1 feedback but not LM to V1 feedback facilitates bursting. Related to the point above, how can they exclude that intrinsic properties of V1 and S1 neurons do not contribute to this difference in spiking? Indeed, strong differences in dendritic integration properties of L5 cells have been reported (Fletcher and Williams 2019, DOI: 10.1016/j.neuron.2018.10.048) along the rostro-caudal axis in the cortex. It seems therefore likely that S1 is more prone to bursting than V1, independently of the difference in the feedback pathways. Moreover, the S1 example cells in Figure 5B and Figure S2 already seem to show a higher probability of producing several spikes than V1 in the 'feed-forward' condition without feedback activation. Careful control experiments are necessary to ascertain that the observed differences in 'bursting' are not due to differences in intrinsic properties of V1 and S1 neurons. Finally, according to Figure 3, layer 5 pyramidal cells in S1 receive stronger connections from M1 than V1 cells from LM. Therefore, the difference in the number of spikes evoked could also be due to a difference in the amount of current induced. Titrating the laser power might provide clarification.

3) Moreover, the difference in spiking evoked by combined feedback and 'feed-forward' stimulation is that M1 to S1 is likely to evoke a third spike, while LM to V1 at most evoke two spikes (the probability of evoking two spikes is similar for the two pathways). To conclude from this relatively subtle difference that specifically only M1 to S1 feedback facilitates bursting is clearly overstated.

4) The authors show that under sustained current injection, optogenetic activation of feedback axons elicits a transient increase followed by a prolonged decrease of the firing rate of pyramidal cells. The authors refer to this effect as temporal sharpening, and seem to claim that this is a specific property of feedback connections. However, the sharp increase in activity followed by suppression described here is a predictable result of strong simultaneous activation of many inputs onto a cortical circuit and likely at least partly due to the recruitment of recurrent inhibition (rather than only due to the direct recruitment of inhibitory neurons by feedback). If the authors claim that the neurons' response to feedback activation is more temporally precise than optogenetic activation of other prominent pathways impinging on the same circuit (e.g. feed-forward or higher-order thalamic input) they need to perform more control experiments to compare the impact of feedback projections to other inputs.

5) One of the main differences described was in the timing of feedback-evoked spiking of pyramidal cells. The authors should therefore quantify in more detail the EPSCs and/or EPSPs in pyramidal cells, particularly in terms of kinetics (rise time, decay time, total charge, time to peak or whichever measures they consider most adapted).

Minor comments:

- The abstract stresses that results were obtained with multiple simultaneous whole-cell recordings. Did the authors characterise the connectivity between simultaneously recorded neurons? Was the strength or probability of connections within the local circuit related to the strength of feedback?

- What was the power of the light used?

- What was the minimum number of L2/3 pyramidal cells recorded and used to normalise the response amplitude? Could the authors please add a histogram or other quantification of these normalising factors for V1 and S1 to estimate their variability?

- Figure 3B and 3C could be presented better. For some data points it is not clear if they belong to B or C and the median marks get slightly lost between the single data points.

- The manuscript is in need of proofreading and some cleaning up of prose. Here is a non-exhaustive list of sentences that need attention:

p2: interneurons in cortical circuits are highly heterogeneous grouped into transcriptomic

p8: were the only neurons in vS1 that spikes were elicited by feedback

p15: despite we simultaneously drove

p16: Different from SOM+ cells, the majority of PV+ cells target cell bodies instead.

p16: timing of activities

p16: excited when feed-forward is driven

p17: using a 14 homeothermic blanket system (what is this 14?)

p18: Using the surgical glue

p18: stuck the skins on the skull

p18: to deliver the expression of ChR2-EYFP

p19: generally followed the protocol in the previous study

p19: thus enhance the responsiveness ChR2 expressing axon terminals

p21: the probability of spiking of the second spike

p21: Second, for those trials that a second spike occurred,

p21: we compared these two values of cells in the two feedback pathways

We thank the reviewers and the editor for their time and consideration of our paper for publication. We also thank the reviewers for their careful assessment of our work and their helpful comments and experimental suggestions. The reviewers found that our study was “novel, important, of broad interest” and “well executed”. The reviewers also raised several issues. Following their advice, we performed additional experiments and new analysis and addressed all technical issues that were brought up. We believe the revised manuscript has improved substantially. Please find below our point-by-point responses (reviewers comments in blue and our response in black). Given the situation with COVID-19 pandemic the revision of the manuscript took much longer than expected.

Reviewer #1 (Remarks to the Author):

GENERAL COMMENTS

This is a very nice study demonstrating that feedback (FB) circuits are not anatomically and functionally homogeneous across the brain. Using optogenetic stimulation of FB from either M1-to-S1 or LM-to-V1 combined with whole-cell patch recordings in S1 and M1, respectively, in mouse cortex, the authors show that these two FB pathways target different cell types and modulate excitatory neuron activity in distinct ways. The results are novel, important, of broad interest and the study is well executed. I have several suggestions for improvement, but overall find this an excellent study.

We thank the reviewer for the positive comments.

MAJOR SPECIFIC COMMENTS

1. Results p. 3 and Fig.1. The authors find distinct feedback (FB) laminar termination patterns in V1 vs. S1. However, based on what we know from primate visual cortex, it is very possible that the LM-to-V1 pattern seen here is unique to FB from V2 to V1, rather than a general rule of FB connectivity within visual cortex. The authors should comment about this in the discussion.

We thank the reviewer for bringing up this point. Indeed, we found that LM to V1 feedback projections did innervate L4 in V1, which seems to be against the canonical notion that feedback connections avoid L4 in other species such as primates (Barone et al. 2000) and rats (T. A. Coogan and Burkhalter 1990; T. a. Coogan and Burkhalter 1993). We also found that activation of feedback elicited responses in L4 neurons, consistent with the L4 projection pattern (Figure 1E, Figure 3B and C). Interestingly, a recent paper revealed that the feedback from the secondary somatosensory cortex (S2) to S1 also sends some sparse projections to L4 (Minamisawa et al. 2018), suggesting a similar feedback projection pattern to L4. Critically, we found that LM to V1 feedback terminals in L4 were sparser than those in other layers. Therefore, in this sense, LM to V1 projection patterns are generally consistent with the classical view of feedback patterns. Further studies on other feedback pathways are needed to determine

how universal the presence of weak L4 feedback projections is between other cortical areas. As requested, we added a paragraph in the discussion to clarify this issue (Page 15, Line 393-400).

2. Fig. 1B. I suggest to lightly outline the areal borders in the optical maps, as this makes it easier to see how the areas were identified in C.

We added the areal borders in the optical maps in Figure 1B as the reviewer suggested.

3. Fig. 1D. The conclusion that M1 to S1 FB connections are less topographically organized than LM-to-V1 seems unwarranted. The comparison should be made for FB terminals within V1 only vs. within S1 only. If the dashed outline in D represents all of S1 then FB to S1 is also somewhat topographically organized, as only a region of S1 shows green label. Also there is no quantitative analysis similar to that done in panel B for the M1-to-S1 projection. This analysis should be done or the claim that the latter FB is less topographic should be dropped.

We agree with the reviewer. We did not provide quantitative evidence that M1 to S1 projections are not topographically organized. Moreover, a previous study suggested topographical organization of feed-forward projections from vS1 to vM1 projections (Mao et al. 2011). The vM1 to vS1 feedback projections may also be topographically organized. In light of this, we revised our statement in the results (Page 3, Line 94-95).

4. Fig. 1E. Some form of staining that demonstrates layers and area specificity should be added next to each micrograph of the fluorescent label.

We thank the reviewer for the suggestion. For layer specificity, we added examples with DAPI staining in Figure 1E, which was helpful in identifying different layers in V1 or vS1 areas. To guarantee area specificity in the virus injection, we injected 2% Fast Green into V1 or vS1 areas, identified either with the intrinsic imaging map (V1) or stereotaxic coordinates before we perfused and sliced the brain. We now further describe the histology procedures in the Methods section (Page 19-20, Line 578-588).

5. Was the location of the injection sites confirmed in histological sections to determine whether the results presented here apply to both FB arising from superficial as well as deep layers? Images of at least some injection sites should be shown as supplementary material.

Discriminating feedback projections from superficial versus deep layers is indeed important. However, we did not aim to address this point in this study. Therefore, in our experiments we tried to cover both superficial layers and deep layers with our injections by targeting two depths, 300 μm and 700 μm relative to the brain surface, described in the Methods section (Page 19, Line 571-573). Before each recording, we confirmed under two photon microscopy that our injection method resulted in abundant fluorescent labeled axon terminals in V1 or vS1

originating from LM and vM1 respectively. In addition, given the reviewer's comment we performed an additional set of experiments injecting two animals in LM and three animals in vM1. We sliced their brains and measured the spread of infection at the injection sites. We now show two examples of the injection sites for each area in the Supplementary materials (Supplementary Figure S1). These results demonstrate that our dual injection method infected both deep and superficial cortical layers.

6. Methods P.19. It is stated that whole-cell patch recordings were performed only in L2/3, 4 and 5. But this is inconsistent with several figures demonstrating also L1 inhibitory cells were recorded from. It is a real shame that there are no L6 cell recordings as this is a major target of FB connections.

First, we thank the reviewer for noticing these discrepancies. We modified the text to clearly define the laminar distributions of our recordings which indeed also included patching in L1 (Page 20, Line 609). We agree that L6 is very interesting especially given that it is a major target of feedback projections. However, due to the cell type heterogeneity of L6 (Gouwens et al. 2019; Scala et al. 2020), achieving a complete picture of connectivity and functional significance requires a significant number of many more experiments which is beyond the scope of the current manuscript. To address this we extended our discussion to acknowledge the importance of feedback projections on L6 and highlighted the need for follow-up studies (Page 15, Line 404-408).

7. Methods P.19 and Fig.S3. The criteria used to determine monosynaptically evoked EPSCs and EPSPs seem somewhat weak. First it is stated that to identify monosynaptic events TTX and 4-AP were applied to the slices in a "subset of the experiments". Please specify what fraction of experiments this was done for.

We thank the reviewer for bringing up this point. We recorded 32 V1 cells (out of 291 cells in total) and 27 vS1 cells (out of 225 cells in total) with TTX and 4-AP. The numbers are now listed in the text (Page 7, Line 165-166).

As a second criterion the authors used 8 ms latency of evoked events as the cutoff latency to identify monosynaptic connections. However, in Fig. S1C the appropriate latency would seem to be <4ms, as this is the only latency that does not show overlap between recoverable and non-recoverable cells. The analysis should be repeated using 4 ms as the cutoff latency.

We are confident that the polysynaptic events were rare in our experimental conditions due to the following reasons: a) we never found a pyramidal cell to fire in our experiments, which indicates that the chance of polysynaptic events are low; b) in the subset of experiments with TTX-4AP applied during the recording, responses of the majority of the cells (38/43) we recorded were recovered in presence of 4-AP.

Our study aims to provide an estimate of the monosynaptic connectivity rate and strength for feedback connections. Based on the reviewer's comments we redid this analysis. We chose a threshold of 7 ms which includes all monosynaptic events based on the TTX-4AP experiments (see figure below) with a minimum number of false positives (i.e. potentially polysynaptic events). Please note that the false positive estimate is based on the TTX-4AP experiment which is a stringent test for monosynaptic events. This is because it is almost impossible for polysynaptic responses to recover, while some small monosynaptic events may fail to recover, because the presence of TTX-4-AP typically causes a significant reduction in the EPSC amplitudes (Cho et al. 2013; Petreanu et al. 2009). In addition, the fact that we never found a pyramidal cell to fire in our experiment provides strong evidence that our monosynaptic connectivity rate estimates are not significantly affected by polysynaptic events. Moreover, the latencies in our study are longer than typical latencies from other studies (Lee et al. 2013). This is because our recordings were performed under room temperature, which preserves the quality of tissues better than using physiological temperature (Ballanyi and Ruangkittisakul 2009; Ting et al. 2014), but leads to larger latencies in synaptic events (Taschenberger and von Gersdorff 2000). We apologize for not making this point clear in the original submission. Now the reasoning is clarified in the result section of connectivity (Page 7, Line 158-171), the Methods section (Page 22, Line 676-681) and Supplementary Materials (Section "Most of the recorded EPSC and EPSP events were monosynaptic").

Figure 3 in revision: Connectivity data reanalyzed with 7 ms as the latency cutoff of presumably monosynaptic events. Panels B and C were also reorganized to violin plots to improve the presentation of the results.

Nevertheless, following the specific suggestion of the reviewer we also checked if a 4 ms threshold changed our results (Supplementary Figure S6). Our findings did not change. The proportions of responsive cells (Figure 3A) became lower, but the conclusions for the relative strength of feedback connections remained the same.

Figure S6 in revision: Connectivity data reanalyzed with 4 ms as the latency cutoff of presumably monosynaptic events.

In panel C what does the Y axis measure? Trace counts? Does this correspond to cell numbers? if not please define it.

Following the reviewer's suggestion we redid this analysis showing the recoverable and non-recoverable cell count distribution of their average latencies across trials (Supplementary Figure S5C).

Figure 5C. Distribution of the average latencies across traces for recoverable cells and unrecoverable cells. We set 7 ms as the latency cutoff for presumably monosynaptic events.

Moreover, what is the 3/83 of recoverable events mentioned in the legend? This number does not seem to agree with any of the figures in panels B-C.

This was a typo. We apologize for the confusion. After we changed Figure S5C from trace counts to cell counts, this number became 5/43, which is referred to in the Supplementary Materials (Page 4, Line 75).

8. What is the LED light intensity used for photostimulation (in mW/mm²), and were control experiments performed in the absence of opsins to determine potential light-induced effects?

It was approximately 6 mW/mm². As a control, we recorded several cells in the cortex in the absence of the opsin, and found that there was no activity in response to the light. We added this information in the Methods section (Page 20, Line 626) and the results of this control experiment are now shown in the Supplementary Materials (Supplementary Figure S4).

9. What was the spatial spread of LED activation? Were the authors activating FB across all layers, or were they able to focally activate the FB in different layers? I am wondering here whether the authors could distinguish between FB projecting to different layers. This question is motivated by the possibility that there exist different sets of FB connections related to different layers that could have different impacts on postsynaptic cells. If the authors did not distinguish between effects of FB to different layers, this should be made clear in the discussion, particularly in the context of predictive coding models, which indeed postulate different FB effects in different layers.

The 470 nm light was delivered through the 40x objective lens perpendicularly to the brain slices with a spatial spread of approximately 1 mm². This covered all layers of the cortex and therefore, we did not distinguish feedback terminals projecting to different layers. We now clarify

this in the method section (Page 20, Line 623-625) and extend our discussion, highlighting the importance of dissecting the feedback terminals residing in different layers in future studies (Page 15-16, Line 408-413). However, we recorded V1 and vS1 neurons in different layers, which serves as the context to discuss different FB effects in different layers.

10. Results pp 7-8 and Fig.3. There seem to be more interesting laminar differences within the same FB pathway than the authors point out. For example, the effects on SOM or excitatory cells in V1 seem to differ in L2/3 vs L5. Any superficial vs deep layer differences should be emphasized, as there are theories that FB to different layers serve different functions. The data in Fig.3D should also be split by layers for the same reason. In this respect it is a shame that no recordings were performed in L6 as this is a major target of FB connections.

We thank the reviewer for bringing up this point. We added a paragraph in the results section to compare the connectivity difference of L2/3 vs L5 across the feedback pathways (Page 7-8, Line 186-200). Figure 3D is already split by layers. We apologize for not marking this clear. We now added the labels for layers at the bottom of the figure. We also changed the connectivity results into violin plots that makes the border between panel B and C clearer. As we discuss above we also agree with the reviewer that analyzing feedback to L6 is important and future work should address this (Page 15, Line 404-407).

11. Results p. 10 and Fig.4. It would be very interesting (and revealing about underlying mechanisms) to know how the inhibitory neurons responded to these experimental manipulations. Didn't the authors record from inhibitory neurons too during these experiments?

We agree with the reviewer that these experiments are of great interest. However, these experiments were originally out of the scope of the current paper. To perform experiments that cover all the cell types and layers, we need to perform experiments on dozens of more mice, which would require a new series of experiments that are beyond the scope of the current work.

12. Discussion p. 15, 2nd paragraph :” Previous studies have revealed that SOM+ cells are involved in surround suppression because their activities increase with the stimulus size...”. But also because the same cited studies showed that inactivation of SOM cells reduces surround suppression.

We edited the sentence in the discussion as the reviewer suggested to make the statement more accurate, as follows:

“Previous studies have revealed an important role of SOM+ cells in surround suppression. Specifically, the activity of SOM+ cells increases with the stimulus size and both activation and inhibition of their activity affects surround suppression in pyramidal cells (Adesnik et al. 2012; Nienborg et al. 2013).” (Page 16, Line 423-425)

13. Discussion p. 15, 2nd paragraph:” Our finding here filled in this gap by showing that feedback from LM to V1 recruits local SOM+ interneurons.”. The authors have not filled this gap, as they have not studied any effects of FB or SOM neuron activity on surround suppression. It would be more accurate here to say something like” our study suggests that the effects of FB on surround suppression are mediated by direct FB contacts with SOM neurons”.

We fixed the statement as the reviewer suggested, as follows:

“In light of these previous findings, our results suggest that the effects of FB on surround suppression may be mediated by direct feedback connections on SOM+ neurons in V1. ” (Page 16, Line 427-428)

14. Discussion p. 16, 2nd paragraph. It is not correct to state that Predictive Coding models view FB as suppressive. The Predictive Coding model proposed by e.g. Friston and colleagues, call for two different kinds of FB. One FB, to L6, is suppressive and driving, the other, to superficial layers, is modulatory and excites similar features but suppresses orthogonal features.

We thank the reviewer for highlighting this issue. We reorganized the paragraph accordingly by explicitly stating the two kinds of feedback.

“ One of the leading models that have been proposed for the feedback integration is the predictive coding model (Rao and Ballard 1999; Bastos et al. 2012; Keller and Mrsic-Flogel 2018). In this model, feedback is interpreted as the prediction signal while the feedward input is the prediction error (Rao and Ballard 1999). With this notion, Friston 2018 (Friston 2018) predicted two different types of feedback projections. One type is to deep layers which is suppressive, and the other is to superficial layers which is modulatory and excites similar features but suppresses orthogonal features. We found that LM to V1 feedback projected to both pyramidal cells and interneurons (i.e. SOM+ cells and PV+ cells) in L2/3 which could serve as a modulatory signal with both excitatory and inhibitory components. In contrast, L5 pyramidal cells mostly received inhibition from feedback, thus being more suppressive, which is consistent with the predictions of the predictive coding model. However, future work is still needed to examine how LM neurons precisely connect to V1 neurons with different functional features and how this process is regulated by interneurons, especially under in vivo conditions. Several recent studies have explored this direction. In particular, Marques et al., 2018 shows that compared to local V1 neurons, LM to V1 feedback boutons over-represent the visual areas perpendicular to their preferred orientation of the boutons, but are aligned with their preferred direction (Marques et al. 2018). Building on the results we show here, it will be intriguing to further dissect the microcircuitry to directly test the predictions for the predictive coding model. ” (Page 17, Line 477-493)

15. The discussion section does not read very well, and needs to be improved in both English as well as content. The section on predictive coding is not well related to the results of this study.

As suggested by the reviewer, we reorganized the whole discussion section, especially the predictive coding part.

MINOR SPECIFIC COMMENTS

1. Please number the pages. It makes it easier for Reviewers.

We added the page number to the documents in the revision.

2. Page 2. "Among GABAergic interneurons,.....GABAergic interneurons". Please specify these facts apply to mouse cortex, as for example it is not known whether these 3 classes of neurons are non-overlapping or whether they represent 80% of GABAergic neurons in primates. Also Ref. 15 is limited to VIP neurons; it would seem more appropriate here to cite a few more comprehensive reviews.

We clarified the reference to the mouse neocortex (Page 2, Line 45) and added more references accordingly (Page 2, Line 47).

3. Fig. S1 legend. Typo "bursty" not "busty".

Fixed (Figure S3 legend).

4. Results. p. 8 line 12. The correct figure citation here is 3D not 3C.

Fixed (Page 8, Line 197).

5. Results. p.9 bottom of page, paragraph title. The term principal cells here is inappropriate. All that you know is that these are excitatory cells, with no knowledge of whether these are principal cells.

We replaced all the terms "principal cells" with "excitatory cells" as suggested by the reviewer.

Reviewer #2 (Remarks to the Author):

Shen and colleagues compared two feedback projections in the neocortex of the mouse. The first is a projection from vibrissal motor cortex (vM1) to barrel cortex (vS1), and the second is a projection from the lateral-medial extrastriate area (LM) to the primary visual cortex (V1). They used optogenetics and patch clamp recordings to map the cell type-specific strength of the feedback input. They find that, while vM1 -> vS1 projections preferably contact vasointestinal

peptide-positive neurons, the LM → V1 projections preferable contact somatostatin-positive cells. They went on to determine the functional consequences. They found that coincident activation of feed-forward input to pyramidal cells in S1, and feedback input from vM1, were more likely to evoke a burst response compared to the same stimulation protocol in the visual area.

Altogether, I did not find this work suitable for Nature Communications for the following reasons:

1) Assuming that the results are accurate, I am not sure whether this work presents a sufficient conceptual advance to warrant publication in Nature Communications. First, the logic, or argument, for comparing these two pathways is not clear. It seems to me that one could have picked any two top-down pathways and compared them, and it would not have surprised me if there were always some differences.

It is also not entirely clear why the authors focused on the differences in connection strength between VIP and SOM neurons. They also found many other differences between other cell types, and these may even be more functionally relevant.

We thank the reviewer for her/his feedback and for the criticism which provides us the opportunity to highlight the novelty of our findings and discuss why we think that our work is appropriate for Nature Communications. Feedback projections between cortical areas are ubiquitous across species and have been hypothesized to be important for numerous brain functions like learning, attention and perceptual inference. However, despite decades of research we are still missing basic information about feedback, such as the basic wiring patterns across different areas which limits a mechanistic understanding of its function. Prior to our study, cortico-cortical feedback projections were generally thought to be disinhibitory, with feedback projections targeting primarily VIP+ interneurons, who inhibit SOM+ interneurons that in turn disinhibit excitatory neurons (Lee et al. 2013; Zhang et al. 2014; Niell and Stryker 2010). Another recent study shows that SOM+ neurons receive relatively weak long-range connections regardless of input areas (Naskar et al. 2021), including vM1 and S2. However, here we found that feedback from LM to V1 targets SOM+ instead of VIP+ interneurons, which is novel (see also comments by reviewer 1 above) and unique among the cell-type connectivity rules of long-range cortical feedback projections reported in the literature. By comparing the results between M1 → S1 and LM → V1 feedback pathways, we not only highlighted the differences in the connectivity, but also reproduced the M1 to S1 pattern that was described in the previous literature (Lee et al. 2013), providing an important replication of this finding and also confirming that the distinct pattern we see in visual cortex is not due to some difference in our experimental approach. Because our results demonstrate that there is not a single universal feedback wiring motif across all cortical areas, they emphasize the importance of further studies to systematically characterize the connectivity diagram of different cortical feedback pathways using more high-throughput methods such as electron microscopy or mesoscale connectome techniques (Huang et al. 2020). We believe the clear view of the diversity in these projections

provided by this work will motivate further studies along these lines. We now emphasize this in the discussion section of the revised manuscript (Page 15, Line 385-392). We also agree with the reviewer that apart from the VIP+ and SOM+ interneurons, differences in connectivity with other cell types should be also more emphasized. Therefore, in the revised manuscript, we now emphasize the differences we found in projections to other neuronal types across cortical layers (Page 7-8, Line 186-200).

2) On the technical side, I am not convinced that the observed differences are not due to the experimental design.

First, the authors elaborated on the issue of polysynaptic activation. While the pyramidal cells do not typically fire when stimulating the feedback projections, the inhibitory cells often do. This may be due to forward inhibition and could affect the amplitude of the excitatory responses in a cell type-specific manner. Examples of this may be seen in Fig.2E; The LM->V1 projections triggers an EPSP that is rapidly followed by an IPSP. This seems not the case for the vM1->S1 projection. While this is, of course, just a single example, it is not clear to what degree this will affect the amplitude of responses and distort the results.

We thank the reviewer for the opportunity to clarify this point. In order to avoid confounds from polysynaptic inhibition distorting our results, we also measured EPSCs in addition to recording EPSPs. EPSCs characterize the strength of synaptic connection from feedback terminals to cells in V1 or vS1, measuring only excitation and not inhibition. Specifically, using current clamp, we minimized the inhibitory current by setting the clamped voltage at the reversal potential of the inhibitory channels, which is about -85 mV. Therefore, the EPSCs we recorded are not likely to be strongly affected by the IPSC. The EPSPs reflect more the overall activity of a neuron evoked by the feedback stimulation, and thus could include some polysynaptic inhibitory events. Using both measures, we were able to estimate the relative strengths of connections onto each cell type (EPSCs measured in voltage clamp) as well as the network effect of the feedback (EPSPs measured in current clamp). We apologize for not making clear the different purposes of the two recording modes. Now in the revised manuscript, we extended the method section accordingly (Page 20, Line 618-622).

Second, and related to the point just above, the full-field blue light stimulation in Figures 4 and 5, may trigger complex interactions between cells in the network. Hence, it is not obvious to relate the functional differences between the feedback projections to the differences in the wiring diagram. Is this really due to the differential input strength to VIP and SOM cells?

We thank the reviewer for bringing up this important point. Indeed, complex interactions between cells due to full-field stimulation make it difficult to interpret the functions of a microcircuit. However, in our experiments the ChR2-expressing cell bodies were excluded from the slices we recorded (see Methods, Page 20, Line 601-604), which avoids light-evoked complex interactions among cells in LM or vM1. Moreover, we never found excitatory neurons in

LM or vM1 to be directly driven to fire by the feedback activation. Therefore, the polysynaptic excitatory interactions in LM or vM1 networks are at best weak. The vast majority of the excitatory events are monosynaptic, which is also supported by our control experiment in the presence of TTX and 4-AP (Figure S5C pasted below, please also refer to the response to Reviewer 1). The polysynaptic events due to activating inhibitory cells are addressed in our study in Figures 2 and 3. We agree with the reviewer that the functional differences between the two feedback pathways we measured (Figure 5) may not necessarily be due to the differential input strength to VIP+ and SOM+ cells alone but a network effect. We now emphasize this point in the discussion section (Page 17, Line 467-469).

Figure 5C. Distribution of the average latencies across traces for recoverable cells and unrecoverable cells. We set 7 ms as the latency cutoff for presumably monosynaptic events.

Third, the authors do not report the light density used to activate the opsin. In particular, for Figures 4 and 5, the light intensity may change the outcome. I would recommend performing a light intensity input-output relationship to see whether conclusions hold for different light intensities.

We agree with the reviewer that light density is a relevant factor affecting our measurements. Additionally, the expression of the virus also affects the strength of activating the feedback. Our goal for the connectivity measurements was to maximize the activation of feedback. To account for the variability across experiments we normalized all measurements relative to feedback -> L2/3 pyramidal cell connectivity (we describe these in Methods section Page 22, Line 685-687). We also reported that we used 6 mW/mm² in the Methods section (Page 20, Line 626)

In our functional experiments (Figures 4 and 5), we adjusted the light intensity to restrict the level of the light-evoked EPSC from 100 pA to 500 pA to facilitate comparisons between experiments. Within this range the differences in effects described in Figures 4 and 5 were consistent across experiments for vM1->vS1 and LM->V1 (e.g. no overlap in the results between the two pathways in Figure 5 E and F). We now clarified this approach in the Methods section (Page 21, Line 656-658).

Fourth, the authors note that the strength of connections from vM1 is generally weaker than the LM projection. The former is, however, a longer projection and opsin expression density at the axon terminals may, therefore, be weaker.

We thank the reviewer for her/his attention to this potential issue. We agree with the reviewer that the opsin expression density could affect the overall strength of the synaptic events recorded. However, for this reason we compare our connectivity results in terms of normalized values, excluding the possibility of the connectivity differences between the two feedback pathways to address this potential experimental design artifact.

3) It is not clear to me whether the stimulation protocols in Figures 4 and 5 bear any relevance for *in vivo* activity. We still know very little about these projections. We do not know when these feedback projections are active during behavior, and we do not know whether these projections fire rate-coded output or temporally precise brief feedback. In addition, the disadvantage of using full-field optogenetic stimulation is that all feedback projections are activated simultaneously, which is a sledge-hammer effect for the network, and which most likely does not bear relevance for how the circuit operates.

We agree with the reviewer that the optogenetic experimental paradigm we used is artificial and the slice preparation has major limitations in deciphering the function of feedback ultimately relating to cortical computation and behavior. Detailed *in vivo* measurements of feedback activity under different stimulation conditions and behaviors are needed to understand how and when feedback is active. Here the primary goal of our study was to characterize the cell type-specific connections of feedback in the cortex, and this approach yielded evidence of distinct wiring between the two feedback pathways we analyzed. Moreover, our work confirmed the existence of a disinhibitory circuit in vS1 and at the same time provided the first evidence that LM to V1 feedback may serve a different function. This result proposes a potential role of feedback that of course needs much more *in vivo* research that is beyond the scope of the current paper. In the revised version of the manuscript, we extend our discussions, including the potential issues raised here, and emphasize the need for future *in vivo* studies to elucidate the function of feedback (Page 17, Line 485-489).

4) The manuscript is not an easy read. I would recommend moving many of the technical details to the methods section to help the reader focus on the concepts and results. Right now, the message is drowning in the technical details.

As suggested by the reviewer, we reorganized the text by moving technical details in the methods section accordingly.

Reviewer #3 (Remarks to the Author):

Using in vitro electrophysiology and optogenetics, Shen and colleagues characterize the connectivity profiles of two intra-cortical feedback pathways. The authors show that projections from primary motor cortex (M1) in primary somatosensory cortex (S1) make stronger connections onto vasointestinal peptide (VIP), while feedback projections from the lateromedial visual area (LM) to primary visual cortex (V1) makes stronger connections onto somatostatin (SOM) positive interneurons. Moreover, M1 projections are less connected with layer 4 cells of their target region. For both pathways, activating feedback projections induces a phase of brief spiking followed by a suppression of activity in the target region. When optogenetic stimulation of feed-back projections is combined with a brief current injection in the target neurons (called incorrectly feed-forward input), M1 input onto S1 neurons is more likely to induce a third spike, while LM input onto V1 neurons induces maximally two spikes, but with less jitter than for M1 to S1 connections.

Numerous types of long range cortico-cortical connections have been described but it remains unclear what rules govern their organisation. In depth characterisation of the different circuit motifs that can be recruited by these projections is therefore important to further our understanding of cortical communication. However, I have major concerns about the interpretation of some key results of this study:

We thank the reviewer for their positive assessment of the importance of the question we are addressing in order to understand cortical communication.

Main concerns:

1) The authors claim that the difference they observe is a general distinction between intra-modality feedback and motor to sensory feedback (page 2 and 15). However, they compare different input areas (M1 and LM) targeting different output areas (S1 and V1). From the presented data alone it is therefore impossible to make this claim. Differences observed could be due to differences in circuit and cellular properties of V1 and S1 rather than due to the modality of the feedback projection (see also point 2). Ideally the authors should compare sensory and motor feedback onto the same target area and neurons, e.g. M1 and S2 onto S1. If this is beyond the scope of this study, at least the authors need to tone down their claims and discuss this issue carefully in the discussion.

We thank the reviewer for this point. Our study focused on characterizing LM to V1 feedback connections and comparing it with established feedback connections principles largely described in other studies (Lee et al. 2013; Zhang et al. 2014). This allowed us to observe that LM to V1 feedback targets L2/3 SOM+ cells instead of VIP+ cells, which is different from reports describing other feedback pathways (Lee et al. 2013; Zhang et al. 2014). We are the first to show that feedback targets SOM+ cells. By comparing our results with the previously well-described feedback connectivity from M1 to S1, we highlighted these differences and used

the M1 to S1 data we collected as a control to confirm the reliability of our experimental conditions. However, we agree with the reviewer that a comparison of feedback connectivity onto V1 neurons from non-visual areas is important. To this end, Yang Dan's lab found that feedback from anterior cingulate cortex also targets VIP+ cells (Zhang et al. 2014). Further studies are needed to reveal all the different cortico-cortical feedback cell type specific motifs and discover the logic governing when these different motifs are applied. We now extend our discussion to explain the rationale behind our approach and highlight the need for additional work and discuss our conclusions in light of the limitations of our study (Page 15, Line 382-392).

2) The authors claim that M1 to S1 feedback but not LM to V1 feedback facilitates bursting. Related to the point above, how can they exclude that intrinsic properties of V1 and S1 neurons do not contribute to this difference in spiking? Indeed, strong differences in dendritic integration properties of L5 cells have been reported (Fletcher and Williams 2019, DOI: 10.1016/j.neuron.2018.10.048) along the rostral-caudal axis in the cortex. It seems therefore likely that S1 is more prone to bursting than V1, independently of the difference in the feedback pathways. Moreover, the S1 example cells in Figure 5B and Figure S2 already seem to show a higher probability of producing several spikes than V1 in the 'feed-forward' condition without feedback activation. Careful control experiments are necessary to ascertain that the observed differences in 'bursting' are not due to differences in intrinsic properties of V1 and S1 neurons. Finally, according to Figure 3, layer 5 pyramidal cells in S1 receive stronger connections from M1 than V1 cells from LM. Therefore, the difference in the number of spikes evoked could also be due to a difference in the amount of current induced. Titrating the laser power might provide clarification.

We agree with the reviewer that it is possible that there is a higher percentage of L5 cells in S1 than in V1 that exhibit bursting. For this reason, we preselected bursty neurons to perform these experiments (refer to Supplementary Figure S3, S11 for characterization of the morphologies and burstiness of the neurons). These neurons, as reported in the literature, are thick-tufted neurons in both V1 (Shai et al. 2015) and vS1 (Oberlaender et al. 2011). To directly compare their burstiness in our experiments, we recorded the firing patterns of the neurons and measured how prone they were to burst with two metrics: minimal current to elicit a burst and burstiness. Burstiness was defined as the firing rate difference of the spikes inside bursts with that out of the burst, normalized by their summation (see Methods, Page 22-23, Line 642-653), following the approach used in Allen Software Development Kit (Allen SDK). We found that there were no significant differences between V1 bursty neurons and vS1 bursty neurons using both metrics, indicating that the difference we found in bursting with feedback stimulation is not a result of the difference in their intrinsic bursting properties (Figure S11, referred in the main text on Page 13, Line 348-355). We also agree with the reviewer that the strength of the feedback activity will also affect the results in bursting. To make the results comparable across experiments, we recorded the light-evoked EPSCs before the bursting protocol and adjusted the light to make EPSC in a range from 100 pA to 500 pA. We clarified the experimental approach

in more detail in the methods section (Page 21, Line 656-658, please also see comments above for Reviewer 2 point 3).

3) Moreover, the difference in spiking evoked by combined feedback and 'feed-forward' stimulation is that M1 to S1 is likely to evoke a third spike, while LM to V1 at most evoke two spikes (the probability of evoking two spikes is similar for the two pathways). To conclude from this relatively subtle difference that specifically only M1 to S1 feedback facilitates bursting is clearly overstated.

We agree with the reviewer that the difference in the number of spikes the feedback facilitated was subtle, but this effect was significant (Figure 5 C, D, E, F). Specifically, we found a significant difference in the change of firing pattern between the two feedback pathways. In the LM to V1 pathway, feedback elicited one reliable spike that was time-locked to the optogenetic stimulation but we rarely saw another subsequent spike. This pattern is consistent with the light-evoked EPSP recording of L5 neurons (Figure 2C) and temporal sharpening experiments (Figure 4A), where there was a sharp window of the excitatory phase followed by a prolonged inhibitory phase. In contrast, in the vM1 to vS1 pathway, feedback triggered additional spikes that fired at more random times relative to the optogenetic stimulation onset. This effect is very different from the spikes elicited in the LM to V1 pathway where spikes were locked to the optogenetic stimulus. These effects were quantified with the probability of second/third spikes with/without feedback stimulation (Figure 5C) and the distribution of the times for the second spikes relative to the optogenetics stimulus onset (Figure 5D-F). These figures and statistical analysis show the significance of these subtle effects. We further clarified these points in the results section (Page 12-13) and modified the discussion accordingly to avoid overstating the results (Page 16, Line 444-453).

4) The authors show that under sustained current injection, optogenetic activation of feedback axons elicits a transient increase followed by a prolonged decrease of the firing rate of pyramidal cells. The authors refer to this effect as temporal sharpening, and seem to claim that this is a specific property of feedback connections. However, the sharp increase in activity followed by suppression described here is a predictable result of strong simultaneous activation of many inputs onto a cortical circuit and likely at least partly due to the recruitment of recurrent inhibition (rather than only due to the direct recruitment of inhibitory neurons by feedback). If the authors claim that the neurons' response to feedback activation is more temporally precise than optogenetic activation of other prominent pathways impinging on the same circuit (e.g. feed-forward or higher-order thalamic input) they need to perform more control experiments to compare the impact of feedback projections to other inputs.

We agree with the reviewer that the temporal sharpening effect exists in different circuits throughout the neocortex (Liu et al. 2010; Swadlow 2003; Wehr and Zador 2003; Wilent and Contreras 2005; Wu et al. 2008; Hasse and Briggs 2017; Crandall et al. 2015; Markopoulos et al. 2012). This study shows that the feedback circuit also shares this functional property in both

LM to V1 and vM1 to vS1 pathways. We did not mean to imply that neurons' response to feedback activation is more temporally precise compared to other pathways. We now make this clear in the results and discussion (Page 10, Line 285-287, Page 16, Line 431-432). However, we believe that the major source of the inhibition component we found in our experiments was the direct recruitment of the inhibitory neurons from activating feedback because we have never observed feedback driving excitatory neurons to fire under our experimental conditions. However, for *in vivo* conditions, the functional significance of feedback excitation is more complicated than what we described in this study and needs additional work. We now include this point in the discussion (Page 17, Line 485-489).

5) One of the main differences described was in the timing of feedback-evoked spiking of pyramidal cells. The authors should therefore quantify in more detail the EPSCs and/or EPSPs in pyramidal cells, particularly in terms of kinetics (rise time, decay time, total charge, time to peak or whichever measures they consider most adapted).

We thank the reviewer for this good suggestion. We now compare the rise time, decay time and time to peak of L5 pyramidal cells in both V1 and vS1 but did not find a significant difference between them. We show the results in Supplementary Figure S12 and refer to the results in the main text (Page 13, Line 355-359). The only difference we find is the timing of feedback-evoked spiking of pyramidal cells (Figures 4 and 5) which may be attributed to differences in the inhibitory circuits but not the difference in the kinetics of light evoked excitatory responses (Page 13, Line 357-359).

Minor comments:

- The abstract stresses that results were obtained with multiple simultaneous whole-cell recordings. Did the authors characterise the connectivity between simultaneously recorded neurons? Was the strength or probability of connections within the local circuit related to the strength of feedback?

The reviewer brings up an interesting question to find the relationship between the strength of feedback and the local connectivity. We are interested in this in future studies but did not address it here.

- What was the power of the light used?

It was approximately 6 mW/mm². We added this information in the methods section (Page 20, Line 626).

- What was the minimum number of L2/3 pyramidal cells recorded and used to normalise the response amplitude? Could the authors please add a histogram or other quantification of these normalising factors for V1 and S1 to estimate their variability?

We recorded 1 or 2 L2/3 pyramidal cells on each slice as the normalizing factor. The variability of the EPSCs and EPSPs of L2/3 pyramidal cells is shown in Supplementary Figure S7.

- Figure 3B and 3C could be presented better. For some data points it is not clear if they belong to B or C and the median marks get slightly lost between the single data points.

We modified the figure accordingly to make it more clear and easy to interpret. In particular, we used violin plots to better illustrate the distribution of the data (Figure 3, Supplementary Figures S6 and S7).

- The manuscript is in need of proofreading and some cleaning up of prose. Here is a non-exhaustive list of sentences that need attention:

p2: interneurons in cortical circuits are highly heterogeneous grouped into transcriptomic

p8: were the only neurons in vS1 that spikes were elicited by feedback

p15: despite we simultaneously drove

p16: Different from SOM+ cells, the majority of PV+ cells target cell bodies instead.

p16: timing of activities

p16: excited when feed-forward is driven

p17: using a 14 homeothermic blanket system (what is this 14?)

p18: Using the surgical glue

p18: stuck the skins on the skull

p18: to deliver the expression of ChR2-EYFP

p19: generally followed the protocol in the previous study

p19: thus enhance the responsiveness ChR2 expressing axon terminals

p21: the probability of spiking of the second spike

p21: Second, for those trials that a second spike occurred,

p21: we compared these two values of cells in the two feedback pathways

We thank the reviewer for pointing out these issues in the text. We have addressed them.

References:

- Adesnik, Hillel, William Bruns, Hiroki Taniguchi, Z. Josh Huang, and Massimo Scanziani. 2012. "A Neural Circuit for Spatial Summation in Visual Cortex." *Nature* 490 (7419): 226–31.
- Ballanyi, Klaus, and Araya Ruangkittisakul. 2009. "Brain Slices." In *Encyclopedia of Neuroscience*, edited by Marc D. Binder, Nobutaka Hirokawa, and Uwe Windhorst, 483–90.

- Berlin, Heidelberg: Springer Berlin Heidelberg.
- Barone, P., A. Batardiere, K. Knoblauch, and H. Kennedy. 2000. "Laminar Distribution of Neurons in Extrastriate Areas Projecting to Visual Areas V1 and V4 Correlates with the Hierarchical Rank and Indicates the Operation of a Distance Rule." *The Journal of Neuroscience: The Official Journal of the Society for Neuroscience* 20 (9): 3263–81.
- Bastos, Andre M., W. Martin Usrey, Rick a. Adams, George R. Mangun, Pascal Fries, and Karl J. Friston. 2012. "Canonical Microcircuits for Predictive Coding." *Neuron* 76 (4): 695–711.
- Cho, Jun-Hyeong, Karl Deisseroth, and Vadim Y. Bolshakov. 2013. "Synaptic Encoding of Fear Extinction in mPFC-Amygdala Circuits." *Neuron* 80 (6): 1491–1507.
- Coogan, T. A., and A. Burkhalter. 1990. "Conserved Patterns of Cortico-Cortical Connections Define Areal Hierarchy in Rat Visual Cortex." *Experimental Brain Research. Experimentelle Hirnforschung. Experimentation Cerebrale* 80 (1): 49–53.
- Coogan, T. a., and A. Burkhalter. 1993. "Hierarchical Organization of Areas in Rat Visual Cortex." *The Journal of Neuroscience: The Official Journal of the Society for Neuroscience* 13 (9): 3749–72.
- Crandall, Shane R., Scott J. Cruikshank, Barry W. Connors, Shane R. Crandall, Scott J. Cruikshank, and Barry W. Connors. 2015. "A Corticothalamic Switch: Controlling the Thalamus with Dynamic Synapses." *Neuron* 86 (3): 1–15.
- Friston, Karl. 2018. "Does Predictive Coding Have a Future?" *Nature Neuroscience*.
- Gouwens, Nathan W., Staci A. Sorensen, Jim Berg, Changkyu Lee, Tim Jarsky, Jonathan Ting, Susan M. Sunkin, et al. 2019. "Classification of Electrophysiological and Morphological Neuron Types in the Mouse Visual Cortex." *Nature Neuroscience* 22 (7): 1182–95.
- Hasse, J. Michael, and Farran Briggs. 2017. "Corticogeniculate Feedback Sharpens the Temporal Precision and Spatial Resolution of Visual Signals in the Ferret." *Proceedings of the National Academy of Sciences*, 201704524.
- Huang, Longwen, Justus M. Kebschull, Daniel Fürth, Simon Musall, Matthew T. Kaufman, Anne K. Churchland, and Anthony M. Zador. 2020. "BRICseq Bridges Brain-Wide Interregional Connectivity to Neural Activity and Gene Expression in Single Animals." *Cell* 182 (1): 177–88.e27.
- Keller, Georg B., and Thomas D. Mrsic-Flogel. 2018. "Predictive Processing: A Canonical Cortical Computation." *Neuron* 100 (2): 424–35.
- Lee, Soohyun, Illya Kruglikov, Z. Josh Huang, Gord Fishell, and Bernardo Rudy. 2013. "A Disinhibitory Circuit Mediates Motor Integration in the Somatosensory Cortex." *Nature Neuroscience* 16 (11): 1662–70.
- Liu, Bao-Hua, Pingyang Li, Yujiao J. Sun, Ya-Tang Li, Li I. Zhang, and Huizhong Whit Tao. 2010. "Intervening Inhibition Underlies Simple-Cell Receptive Field Structure in Visual Cortex." *Nature Neuroscience* 13 (1): 89–96.
- Mao, T., D. Kusefoglou, B. M. Hooks, D. Huber, L. Petreanu, and K. Svoboda. 2011. "Long - Range Neuronal Circuits Underlying the Interaction between Sensory and Motor Cortex." *Neuron* 72 (1): 111–23.
- Markopoulos, Foivos, Dan Rokni, David H. Gire, and Venkatesh N. Murthy. 2012. "Functional Properties of Cortical Feedback Projections to the Olfactory Bulb." *Neuron* 76 (6): 1175–88.
- Marques, Tiago, Julia Nguyen, Gabriela Fioreze, and Leopoldo Petreanu. 2018. "The Functional Organization of Cortical Feedback Inputs to Primary Visual Cortex." *Nature Neuroscience* 21 (5): 757–64.
- Minamisawa, Genki, Sung Eun Kwon, Maxime Chevée, Solange P. Brown, and Daniel H. O'Connor. 2018. "A Non-Canonical Feedback Circuit for Rapid Interactions between Somatosensory Cortices." *Cell Reports* 23 (9): 2718–31.e6.

- Naskar, Shovan, Jia Qi, Francisco Pereira, Charles R. Gerfen, and Soohyun Lee. 2021. "Cell-Type-Specific Recruitment of GABAergic Interneurons in the Primary Somatosensory Cortex by Long-Range Inputs." *Cell Reports* 34 (8): 108774.
- Niell, Christopher M., and Michael P. Stryker. 2010. "Modulation of Visual Responses by Behavioral State in Mouse Visual Cortex." *Neuron* 65 (4): 472–79.
- Nienborg, Hendrikje, Andrea Hasenstaub, Ian Nauhaus, Hiroki Taniguchi, Z. Josh Huang, and Edward M. Callaway. 2013. "Contrast Dependence and Differential Contributions from Somatostatin- and Parvalbumin-Expressing Neurons to Spatial Integration in Mouse V1." *The Journal of Neuroscience: The Official Journal of the Society for Neuroscience* 33 (27): 11145–54.
- Oberlaender, Marcel, Zimbo S. R. M. Boudewijns, Tatjana Kleele, Huibert D. Mansvelder, Bert Sakmann, and Christiaan P. J. de Kock. 2011. "Three-Dimensional Axon Morphologies of Individual Layer 5 Neurons Indicate Cell Type-Specific Intracortical Pathways for Whisker Motion and Touch." *Proceedings of the National Academy of Sciences of the United States of America* 108 (10): 4188–93.
- Rao, R. P., and D. H. Ballard. 1999. "Predictive Coding in the Visual Cortex: A Functional Interpretation of Some Extra-Classical Receptive-Field Effects." *Nature Neuroscience* 2 (1): 79–87.
- Scala, Federico, Dmitry Kobak, Matteo Bernabucci, Yves Bernaerts, Cathryn René Cadwell, Jesus Ramon Castro, Leonard Hartmanis, et al. 2020. "Phenotypic Variation of Transcriptomic Cell Types in Mouse Motor Cortex." *Nature*, November. <https://doi.org/10.1038/s41586-020-2907-3>.
- Shai, Adam S., Costas A. Anastassiou, Matthew E. Larkum, and Christof Koch. 2015. "Physiology of Layer 5 Pyramidal Neurons in Mouse Primary Visual Cortex: Coincidence Detection through Bursting." *PLoS Computational Biology* 11 (3): e1004090.
- Swadlow, Harvey a. 2003. "Fast-Spike Interneurons and Feedforward Inhibition in Awake Sensory Neocortex." *Cerebral Cortex* 13 (1): 25–32.
- Taschenberger, H., and H. von Gersdorff. 2000. "Fine-Tuning an Auditory Synapse for Speed and Fidelity: Developmental Changes in Presynaptic Waveform, EPSC Kinetics, and Synaptic Plasticity." *The Journal of Neuroscience: The Official Journal of the Society for Neuroscience* 20 (24): 9162–73.
- Ting, Jonathan T., Tanya L. Daigle, Qian Chen, and Guoping Feng. 2014. "Acute Brain Slice Methods for Adult and Aging Animals: Application of Targeted Patch Clamp Analysis and Optogenetics." *Patch-Clamp Methods and Protocols* 1183: 221–42.
- Wehr, Michael, and Anthony M. Zador. 2003. "Balanced Inhibition Underlies Tuning and Sharpens Spike Timing in Auditory Cortex." *Nature* 426 (6965): 442–46.
- Wilent, W. Bryan, and Diego Contreras. 2005. "Dynamics of Excitation and Inhibition Underlying Stimulus Selectivity in Rat Somatosensory Cortex." *Nature Neuroscience* 8 (10): 1364–70.
- Wu, Guangying K., Robert Arbuckle, Bao-Hua Liu, Huizhong W. Tao, and Li I. Zhang. 2008. "Lateral Sharpening of Cortical Frequency Tuning by Approximately Balanced Inhibition." *Neuron* 58 (1): 132–43.
- Zhang, S., M. Xu, T. Kamigaki, J. P. Hoang Do, W-C Chang, S. Jenvay, K. Miyamichi, L. Luo, and Y. Dan. 2014. "Long-Range and Local Circuits for Top-down Modulation of Visual Cortex Processing." *Science* 345: 660–65.

REVIEWER COMMENTS

Reviewer #2 (Remarks to the Author):

I appreciate the author's improvements to the manuscript. However, my initial main concern still stands, which is the lack of conceptual advance. Specifically, is it suitable for this journal to compare two very different feedback circuits and find differences between them? The findings contribute to our incomplete understanding of cortical circuit organisation but do not reveal novel general principles or deviations from accepted general principles. Moreover, in my opinion, it is even conjectural whether these circuits are feedback circuits.

In their rebuttal, the authors claim that "prior to our study, cortico-cortical feedback projections were generally thought to be disinhibitory...". Indeed, a few studies have highlighted this by singling out this specific disinhibitory circuit. But this neglects the work showing the excitatory top-down connections onto the tuft dendrites of pyramidal neurons.

My technical concerns were, however, sufficiently addressed.

Reviewer #3 (Remarks to the Author):

I am satisfied with the authors' response to my comments apart from their response to my point 2 (similar to point 3 of reviewer 2). The authors state that they adjusted LED power to achieve EPSCs in the range of 100 to 500 pA. However, this is a large range and this procedure therefore does not completely assuage my concerns. Can the authors please add a figure in the supplementary material quantifying and comparing the strengths of EPSCs evoked by the LED in V1 and S1 cells recorded for the data presented in Figure 4 and 5, to show that there was no systematic difference in how much current was evoked in V1 vs S1 cells.

We thank the reviewers for their positive evaluation of our revision and their constructive suggestions. Here are our point-to-point responses.

Reviewer #2 (Remarks to the Author):

I appreciate the author's improvements to the manuscript. However, my initial main concern still stands, which is the lack of conceptual advance. Specifically, is it suitable for this journal to compare two very different feedback circuits and find differences between them? The findings contribute to our incomplete understanding of cortical circuit organisation but do not reveal novel general principles or deviations from accepted general principles. Moreover, in my opinion, it is even conjectural whether these circuits are feedback circuits.

In their rebuttal, the authors claim that "prior to our study, cortico-cortical feedback projections were generally thought to be disinhibitory...". Indeed, a few studies have highlighted this by singling out this specific disinhibitory circuit. But this neglects the work showing the excitatory top-down connections onto the tuft dendrites of pyramidal neurons.

My technical concerns were, however, sufficiently addressed.

We are glad that our revision effectively addressed the technical concerns of the reviewer. On the conception level, we would like to clarify that our statement in the last-round response "prior to our study, cortico-cortical feedback projections were generally thought to be disinhibitory..." is well aligned with the work of "the excitatory top-down connections onto the tuft dendrites of pyramidal neurons". In fact, the excitatory top-down connections are an important part of the disinhibitory circuits. We have emphasized this relationship in the discussion section:

"VIP+ cells inhibit SOM+ cells that have been shown to target and gate the activity on the apical dendrites⁷⁶⁻⁷⁸. Therefore, without the inhibition from SOM+ in the vM1 to vS1 feedback pathway, the input on the apical dendrites from the feedback projections may induce plateau potentials, propagated to the soma to elicit bursting^{54,55}. Previous work revealed that feedback from vM1 to vS1 elicits calcium spikes at the apical dendrites⁷⁹, and simultaneous activation of the apical and somatic compartments elicit bursting in L5 IB neurons⁵⁴".

Reviewer #3 (Remarks to the Author):

I am satisfied with the authors' response to my comments apart from their response to my point 2 (similar to point 3 of reviewer 2). The authors state that they adjusted LED power to achieve EPSCs in the range of 100 to 500 pA. However, this is a large range and this procedure therefore does not completely assuage my concerns. Can the authors please add a figure in the supplementary material quantifying and comparing the strengths of

EPSCs evoked by the LED in V1 and S1 cells recorded for the data presented in Figure 4 and 5, to show that there was no systematic difference in how much current was evoked in V1 vs S1 cells.

We thank the reviewer for the positive feedback on our revision. It is a good point to examine the light-evoked EPSCs across different areas. To clarify, we only controlled the LED power to achieve a range of 100 to 500 pA in the bursting experiment (Figure 5). We did not control the light intensity in the sharpening experiment (Figure 4). Since we observed highly similar behaviors of cells across areas in Figure 4, this was less of a concern. In this revision, we added Figure S12 showing that the light evoked EPSCs were similar across areas.

Figure S12. Light-evoked EPSCs of V1 and vS1 intrinsic bursty neurons. The bars mark the median values.